# Barriers to Immunotherapy in Ovarian Cancer: Metabolic, Genomic, and Immune Perturbations in the Tumour Microenvironment

**DOI:** 10.3390/cancers13246231

**Published:** 2021-12-11

**Authors:** Racheal Louise Johnson, Michele Cummings, Amudha Thangavelu, Georgios Theophilou, Diederick de Jong, Nicolas Michel Orsi

**Affiliations:** 1Department Gynaecological Oncology, St. James’s University Hospital, Leeds LS9 7TF, UK; amudhathangavelu@nhs.net (A.T.); georgios.theophilou@nhs.net (G.T.); diederick.dejong@nhs.net (D.d.J.); 2Leeds Institute of Medical Research, St. James’s University Hospital, Leeds LS9 7TF, UK; medmic@leeds.ac.uk (M.C.); N.M.Orsi@leeds.ac.uk (N.M.O.)

**Keywords:** ovarian cancer, tumour microenvironment, immunotherapy, resistance, metabolism, innate, adaptive, genomic

## Abstract

**Simple Summary:**

Immunotherapy hinges on stimulating patients’ immune system to fight cancer. This treatment has led to improved survival in patients with malignancies such as melanoma and lung cancer but, disappointingly, its benefits have not been as forthcoming in ovarian cancer. This review summarises how the ovarian cancer tumour microenvironment hinders the efficacy of immunotherapy by modulating immunoregulatory pathways, reorchestrating metabolism and featuring specific cancer cell genomic aberrations. The impact of novel targeted drugs and combination therapies aiming to overcome these obstacles and improve the clinical success of immunotherapy in ovarian cancer are considered.

**Abstract:**

A lack of explicit early clinical signs and effective screening measures mean that ovarian cancer (OC) often presents as advanced, incurable disease. While conventional treatment combines maximal cytoreductive surgery and platinum-based chemotherapy, patients frequently develop chemoresistance and disease recurrence. The clinical application of immune checkpoint blockade (ICB) aims to restore anti-cancer T-cell function in the tumour microenvironment (TME). Disappointingly, even though tumour infiltrating lymphocytes are associated with superior survival in OC, ICB has offered limited therapeutic benefits. Herein, we discuss specific TME features that prevent ICB from reaching its full potential, focussing in particular on the challenges created by immune, genomic and metabolic alterations. We explore both recent and current therapeutic strategies aiming to overcome these hurdles, including the synergistic effect of combination treatments with immune-based strategies and review the status quo of current clinical trials aiming to maximise the success of immunotherapy in OC.

## 1. Introduction

Epithelial ovarian cancer (OC) is the most fatal gynaecological malignancy with around 7400 new diagnoses per year in the United Kingdom alone, and a 5-year overall survival (OS) rate of only 45% [1], prompting efforts towards developing novel treatments. One such area is the rise of immunotherapy as a new therapeutic strategy that is being deployed across a broad spectrum of malignancies.

The tumour microenvironment (TME) is composed of malignant cells, tumour vasculature, lymphatic vessels, fibroblasts and immune cells, including lymphocytes, neutrophils, macrophages, myeloid derived suppressor cells (MDSCs) and dendritic cells (DCs). Tumours can reorchestrate this arrangement to facilitate tumourigenesis, promote angiogenesis and enable metastasis [2]. Cancer cell interaction with host immune cells organises a supportive but exceedingly immunosuppressive TME, allowing cancer cells to evade detection and destruction by the immune system and assisting tumour survival and progression. The adaptive immune response is mainly comprised of CD8+ T-cells, which can target and kill cells altered by infection or cancer; it is this specific cytotoxic response that makes them an attractive target for immunotherapy. T-cells express checkpoint proteins that suppress their activation and function. Inhibiting these proteins with immune checkpoint blockade (ICB) can amplify the anti-tumour T-cell response within the TME to reduce tumour burden. The pioneering work supporting this novel therapy received recognition when awarded the Nobel prize for medicine in 2018 and has led to Food and Drug Administration (FDA) approval for ICB therapy in a variety of tumours, with impressive, robust responses particularly noted in melanoma and lung cancer [3,4].

Other immune strategies have also been deployed as anti-cancer therapy, notably in the context of antigen-presenting cells (APCs). Tumour antigens are processed by APCs, including DCs and macrophages, which present them to T-cells to stimulate a cytotoxic, anti-tumour T-cell reaction within the TME. The FDA approval of a therapeutic cancer vaccine, which enhanced tumour antigen presentation and improved survival in metastatic hormone refractory prostate cancer patients was hailed as a monumental achievement for immunotherapy in this regard [5]. Similarly, modified T-cell transfusions with chimaeric antigen receptor (CAR) have been shown to enhance the immune system’s recognition of tumour specific antigens and gained FDA approval for the treatment of B-cell lymphoma [6]. Anti-tumour vaccines and T-cell therapy are now also being investigated in OC as well as other treatment strategies involving other innate immune system cells.

Despite these early successes, the efficacy of immunotherapy is limited, and many patients develop resistance to therapy. The mechanisms of tumour immune resistance are complex and involve cell autonomous (genomic) mechanisms as well as the modulation of the TME via the suppression of anti-tumour immune response pathways, abnormal neovascularisation and altered metabolism. Restricted therapeutic responses to ICB are observed in OC; they are characterised by poor durability and yet are not without treatment toxicity [7]. Treatment failure in OC is multifactorial. Late diagnosis presents with a greater tumour burden that has fostered a more robust immunosuppressive TME. Metabolic perturbations also have a role to play. As immune cell differentiation and function require energy, the competing metabolic demands of both tumour and stromal cells concurrently impair effector T-cell function by overexposure to suppressive metabolites and essential nutrient starvation [8]. As a result, both innate and adaptive immune responses are blunted, which amplifies tumour tolerance, requiring novel approaches to circumvent these barriers by using alternative immune strategies or combinatorial approaches. Here, we review how genomic, metabolic, and immune factors within the ovarian cancer TME synergistically interact to weaken the anti-tumour response, impairing the effectiveness of ICB and other immunotherapies. We discuss novel strategies designed to overcome these hurdles and summarise ongoing clinical trials aiming to apply immunotherapy in the context of OC management.

## 2. Immunoregulatory Pathways within the TME

The evolution of the immune system is centred on pathogen surveillance and eradication whilst minimising host collateral damage. This is mediated by innate components which distinguish between self and non-self, in addition to immune checkpoints. Together, they regulate the magnitude of the adaptive immune response, thereby generating central tolerance [9]. The first line of host defence is the innate immune system which can rapidly identify cells altered by damage, infection or malignancy through pattern recognition receptors. Once identified, the innate system can deploy phagocytic cells to destroy these abnormal cells, producing a targeted cytokine-mediated inflammatory response and triggering the adaptive immune system. Adaptive immunity refers to a directed, antigen-specific defence mechanism. Major histocompatibility protein complexes I (MHC-I) and II (MHC-II) are responsible for presenting peptide antigens on their cell surface to T-cells; this enables ‘self’ recognition to prevent autoimmunity. An antigen that is recognised as ‘non-self’ via antigen-presenting cells (APC) and co-stimulatory signals (e.g., interleukin (IL)-2, IL-4 and IL-7) [10] triggers an immune response. While immunogenic MHC-I peptides are present on all nucleated cells and recognised by CD8+ T-cells, MHC-II peptides are found on cells such as macrophages and DCs and are recognised by CD4+ T helper (Th) cell subtypes (Th1, Th2, Th17), eliciting the production of cytokines to induce effector T-cell differentiation. This process is co-opted for cancer cell clearance. Tumour-associated antigens presented to T-cells stimulate proliferation and cytotoxic CD8+ T-cell killing of cancer cells within the TME. As this potent CD8+ T-cell reaction has been the principal target for cancer immunotherapy, this review will cover the adaptive immune response first.

### 2.1. The Adaptive Immune Response

The adaptive immune system governs a tailored pathogen response, with safeguards to regulate/suppress the immune response to prevent auto-reactivity. Co-inhibitory molecules such as cytotoxic T lymphocyte antigen 4 (CTLA-4) and programmed cell death protein 1 (PD-1) are induced on T-cell activation and act as physiological brakes on cytotoxic T-cell function to obviate the activity of potentially autoreactive T-cells. Given that CTLA-4 and PD-1/PD-L1 pathways create a mechanism through which cancer cells avoid anti-tumour responses, the blockade of these pathways aimed at reversing T-cell exhaustion, and reinvigorating the anti-cancer response has been central to the development of novel therapies.

#### 2.1.1. CTLA-4

The CTLA-4 receptor ligands CD80/86 are expressed on the surface of APCs and also act as ligands for the co-stimulatory receptor CD28 [11]. Both CTLA-4 and CD28 are expressed on CD4+ and CD8+ T-cells but mediate opposing immunoregulatory functions. CD80/86 interaction with CD28 induces T-cell stimulation but, by contrast, inhibits T-cell responses with CTLA-4 [11]. Although the mechanism is not fully understood, it is thought that CTLA-4 competes with CD28 for ligand binding.

CTLA-4 participates in other aspects of immune control. The subset of CD4+ T-cells that co-express the transcription factor FoxP3 and IL-2 are described as regulatory T-cells (Tregs). Murine models have indicated that their absence causes profound autoimmunity, highlighting their role in T-cell tolerance to self-antigens [12]. Tregs constitutively express CTLA-4, which has a high affinity for CD80/86 ligands. CTLA-4-expressing Tregs capture ligands from opposing APCs cells by a process of trans-endocytosis, thereby acting as a competitive binder for CD28 and thus reducing co-stimulation of CD28 in APCs [13]. In normal tissue, this ensures appropriate immune recognition, but in the TME, an abundance of Tregs leads to the suppression of T-cell mediated immune responses, thereby promoting cancer progression. Moreover, an increased effector CD8+ T-cell:Treg ratio within the tumour infiltrating lymphocyte (TIL) population has been shown to be a crucial predictor of anti-CTLA-4’s clinical efficacy in melanoma patients [14]. Disappointingly, the anti-CTLA-4 treatment’s effects on Treg cells remain inconclusive: While a reduction in tumour-infiltrating Tregs has been observed in some melanoma patients [15], other studies on melanoma, prostate and bladder tumours have reported no such depletion [16]. In OC, low levels of FoxP3+ Tregs and high levels of CD8+ TILs are beneficial to survival [17], suggesting that patients could benefit from immunomodulatory strategies. Indeed, targeted immunotherapy to block CTLA-4 signalling can enhance T-cell activation, shifting the balance away from an immunosuppressive TME. Ipilimumab is the first and only FDA approved CTLA-4 inhibitor after phase III trials demonstrated prolonged survival in metastatic melanoma [18]. However, as of yet there has been no clinical translation of anti-CTLA-4 monotherapy in the OC setting, as reflected by the fact that 95% of OC patients failed to complete a phase II trial due to toxicity, disease progression or death (NCT 01611558) [19].

#### 2.1.2. PD-1

PD-1 is another immune checkpoint protein expressed on the surface of activated CD8+ T-cells, B-cells, macrophages and Tregs, so its selective targeting has broader clinical effects than that of CTLA-4. PD-1 binds to its ligand PD-L1, which is expressed on haemopoietic cells as well as placenta, pancreatic islets and cancer cells, underscoring its purported immunoregulatory role. The PD-1/PD-L1 pathway promotes a hyporesponsive T-cell response, resulting in anergy (diminished antigen response) and apoptosis of CD8+ T-cells, thereby protecting cancer cells from direct cytotoxic attack [20]. Tregs also express PD-1, with ligand binding inducing Treg proliferation, which exacerbates the suppression of TME anti-tumour responses [20]. In this regard, the PD-1 inhibitors nivolumab, pembrolizumab, cemiplimab, dostarlimab and the PD-L1 inhibitors atezolizumab, avelumab and durvalumab have been authorised by the FDA for the treatment of melanoma, non-small-cell lung cancer, renal cell carcinoma, Hodgkin’s lymphoma, head and neck squamous cell carcinoma, urothelial carcinoma and triple-negative breast cancer.

#### 2.1.3. Predictors of ICB Response

PD-1 blockade with pembrolizumab has been approved for the management of any human carcinoma with high microsatellite instability (MSI)/mismatch repair-deficiency (MMR-d) due to its well-established value as a predictive biomarker of immunotherapy response [21]. High MSI/MMR-d causes an accumulation of somatic mutations leading to increased tumour mutational burden and expression of immunogenic tumour neoantigens. The presence of CD8+ TILs, the expression of tumour immunogenic neoantigens and PD-L1/PD-1 positivity are characteristic of the high-MSI/MMR-d phenotype, identifying patients who could benefit from immunotherapy [22]. In 2021, the FDA approved the anti-PD-1 antibody dostarlimab for the treatment of advanced MMR-d endometrial cancer following a successful clinical trial [23]. Unfortunately, MSI or MMR-d only occurs in around 12% of OC and is rare in high-grade serous OC, the commonest histological subtype [24]. Biomarkers such as TILs, immune checkpoint expression and mutational burden could instead be favoured for patient stratification in OC. Disappointingly, early phase clinical trials of anti-PD-1 and anti-PD-L1 monotherapy for OC had limited success with a 9.6–22.2% objective response rate (ORR) [25,26,27,28], prompting the consideration of dual ICB therapy. This has been successful in other malignancies: nivolumab and ipilimumab combination therapy is approved for advanced melanoma after superior survival outcomes compared to ipilimumab monotherapy (2-year OS 63.5% versus 53.6%, respectively), albeit with greater treatment-related grade 3–4 adverse events (55% versus 20%, respectively) [29]. In recurrent or refractory OC, dual ICB therapy offered minor prognostic benefits: Combining nivolumab and ipilimumab led to a limited improvement of median progression-free survival (PFS) compared to nivolumab alone (3.9 versus 2 months, respectively). It did, however, result in a noteworthy increased rate of serious adverse events (>grade 3) in the combination group (49% versus 33%) [30]. Another trial of patients with advanced solid tumours including triple-negative breast, lung, gastric, pancreatic and oesophageal cancers also reported a serious adverse event rate of 73.5% (NCT02658214) [31].

Modifying current ICB techniques to improve their therapeutic benefit has recently been explored in OC. In this regard, current anti-PD-1 antibodies are not optimised to prevent interaction between PD-1 and its alternative ligand PD-L2. The binding of PD-L2 with PD-1 inhibits T-cell CD28-mediated proliferation and CD4+ T-cell cytokine production, further contributing to immunosuppression [32]. In this vein, a retrospective analysis of tumour samples (oesophageal/gastric carcinomas and glioblastoma) from patients with a poor clinical response to anti-PD-1 antibody therapy demonstrated high tumour PD-L2 expression. By contrast, ICB-sensitive bladder cancers have reportedly low PD-L2 expression, implicating PD-L2 as a biomarker of anti-PD-1 clinical response. In OC, both PD-L2 and PD-L1 expression profiles are significantly elevated, making it essential to block both ligands in order to achieve a robust anti-tumour immune response [33]. Recent preclinical studies using an engineered PD-1 receptor with enhanced binding to both PD-L1 and PD-L2 ligands have demonstrated improved anti-PD-1 activity both in vitro in OC cell lines and in vivo in OC mouse models [33]. The potential of this novel approach remains to be fully validated in the context of OC clinical trials.

While much emphasis has been placed on patient hyporesponsiveness to ICB, reports of rapid tumour progression following ICB monotherapy have remained an ongoing clinical concern. Hyperprogression is defined as a ≥2-fold increase in tumour size within a 2-month period of commencing treatment and is associated with markedly worse survival. This pattern of hyperprogression has been reported in a subset of patients with 21 types of cancer who received anti-PD-1 therapy, including OC (12/131), with such events associating with older age (>65 years) [34]. The molecular profiling of another 6 patients (with bladder or lung cancer, endometrial sarcoma or triple-negative breast cancer) with hyperprogression in response to anti-PD-1 and anti-CTLA-4 therapy was associated with *MDM2/MDM4* amplifications (all cases) and epidermal growth factor receptor (EGFR) mutations (2 patients) [35]. MDM proteins are negative regulators of the tumour-suppressor p53 protein (see later) [36], while EGFR activation is associated with the upregulation of the tumour PD-1/PD-L1 pathway, which can drive immunosuppression [37]. Both of these mechanisms can explain the role of MDM2/4 and EGFR in ICB resistance, but the link—if any—to their role in hyperprogression remains unclear. Plausibly, this may reflect the fact that disease progression may have occurred regardless of ICB intervention. However, urgent research is needed to identify the genetic alterations associated with hyperprogression so as to safely align patients with the most appropriate therapy.

#### 2.1.4. ICB Combination Therapies

The lack of durable response with either ICB monotherapy or dual therapy has prompted investigations to focus on a combinatorial approach with conventional OC treatments and immunomodulators. Cytotoxic chemotherapy induces direct cancer cell autophagy, causing the release of immunostimulatory molecules such as lysosomal ATP, which promotes DC recruitment to the TME. Furthermore, chemotherapy-induced cancer cell DNA damage leads to the accumulation of aberrant nucleic acids. These stimulate innate immune signalling through the cyclic guanosine monophosphate-adenosine monophosphate synthase and stimulator of interferon genes (cGAS-STING) pathway and toll-like receptors (TLR 9 or 3), resulting in increased Type I interferon (IFN) production. Together, these mechanisms help to promote DC-mediated presentation of tumour antigens to CD8+ T-cells so as to eliminate residual cancer cells [38]. Additionally, lymphodepletion after chemotherapy mediates an acute state of lymphopenia-induced T-cell proliferation [39]. Furthermore, agents such as cyclophosphamide have been shown to deplete tumour-infiltrating Treg numbers and their suppressive function in mouse models, as demonstrated by a significantly decreased expansion of CD4+/CD8+ T-cells in the presence of untreated Tregs [40].

Evidence of the immunostimulatory effects of chemotherapy has thus provided a rationale for its combination with ICB therapy with a view to further enhance cytotoxic T-cell activity and improve clinical outcomes. Regrettably, phase III trials failed to demonstrate any survival benefit when combining the anti-PDL-1 avelumab with pegylated liposomal doxorubicin chemotherapy in recurrent OC [41]. Another trial was terminated due to lack of efficacy when combining ICB with carboplatin and paclitaxel in untreated, advanced OC (NCT02718417) [42]. However, a subgroup of these patients with tumours positive for PD-L1, CD8+ infiltrates or both exhibited a significant survival benefit with combination treatment [41]. Pembrolizumab monotherapy for advanced OC also resulted in a greater ORR in patients with PD-L1 expression, identifying a subpopulation for future ICB OC studies [43].

The combination of ICB and poly (ADP-ribose) polymerase (PARP) inhibitor (PARPi) therapy is under investigation. PARP is involved in the repair of single-strand DNA breaks through the base excision repair pathway. PARPis lead to the trapping of PARP proteins at sites of single-strand breaks, allowing them to persist unrepaired during DNA replication. Subsequently, this causes the accumulation of double-strand DNA breaks. *BRCA1* and *BRCA2* proteins help repair DNA double-strand breaks via homologous recombination repair (HRR) [44]. As such, the accumulation of DNA damage induced by PARPis selectively kills *BRCA* mutated/silenced OC cells. The effectiveness of PARPis has led to their approval as first [45] or second line [46] maintenance therapy in *BRCA* mutant OC. High-grade serous ovarian cancers (HGSOCs) which are *BRCA* mutant/HRR deficient show increased CD8+ TILs, elevated levels of PD-1/PD-L1 expression and greater neoantigen load, indicating that certain OC subtypes may benefit from PD-1/PD-L1 blockade. PARPi therapy could thus have a potential role in supplementing ICB therapy in *BRCA* mutated OC [47]. Preclinical murine models of *BRCA* deficient OC revealed a strong anti-tumour immune response to PARPi combined with anti-PD-1 [48] or anti-CTLA-4 therapy [49]. Early phase trials combining the anti-PD-1 pembrolizumab and PARPi niraparib in recurrent OC demonstrated a 45% ORR and 73% disease control rate (DCR) in *BRCA* mutated patients [50]. In line with these promising results, anti-PD-L1 durvalumab and PARPi olaparib combination therapy resulted in a 12-week DCR of 81% and 2-year OS of 87% [51]. In addition, the DNA damage induced by PARPi treatment triggers the innate immune response via the production of type 1 interferon via the cGAS-STING pathway [44]. STING signalling promotes TIL recruitment, thereby stimulating anti-tumour immunity. This response can occur independently of *BRCA* status. This was observed in vivo in murine OC models, extending the immunostimulatory value of PARPis beyond *BRCA* mutated patients [52]. In this respect, PARPis have been approved for maintenance therapy of platinum-sensitive OC patients, irrespective of *BRCA* status [53]. Further trials exploring the merits of PARPi combination therapy are ongoing. The phase III ATHENA trial will evaluate anti-PD-1 nivolumab and PARPi rucaparib as maintenance therapy following surgical cytoreduction and platinum-based chemotherapy in advanced platinum-sensitive OC (NCT03522246) [54]. Investigations have been extended to CTLA-4 blockade: An early phase trial recruiting *BRCA* mutated recurrent OC patients will assess the efficacy of the anti-CTLA-4 antibody tremelimumab and olaparib combination (NCT02571725) [55]. PARPis used in combination with ICB could enhance immunotherapy for OC patients and potentially improve survival outcomes.

Other OC treatments have been trialled in the context of immunotherapy combination strategies. The family of vascular endothelial growth factor (VEGF) receptors contains key neoangiogenic regulators whose stimulation promotes endothelial survival, migration, and permeability [56]. As such, VEGF expression is associated with aggressive tumour growth and poor survival in OC [57]. Inhibiting VEGF signalling with the recombinant humanised monoclonal anti-VEGF antibody bevacizumab has been shown to reduce tumour growth and is approved in OC in combination with standard carboplatin and paclitaxel chemotherapy in light of an increased PFS resulting from this regimen [58]. Abnormal tumour vasculature also promotes an immunosuppressive TME characterised by hypoxia and an allied low pH. In particular, VEGF-A promotes Treg proliferation [59] and enhances PD-1 expression on CD8+ T-cells within the TME in mouse models of colorectal cancer, providing a rationale for combining antiangiogenic treatment with ICB [60]. The synergistic effects of bevacizumab and cyclophosphamide with the anti-PD-1 pembrolizumab demonstrated a clinical benefit in 95% (47.5% partial and 47.5% complete response) of patients with recurrent OC in a single arm trial, and the therapy was well tolerated. Moreover, there was evidence of durable response (PFS > 12 months) in 25% of patients [61]. The most common grade 3 events of this triplet combination were hypertension (15%) and lymphopenia (7.5%). Although this was a single-arm study, this strong clinical response highlights the merit of adopting a multi-drug approach to complement ICB therapy. In this respect, the phase III DUO-O trial will investigate the benefit of anti-PD-1 durvalumab in combination with chemotherapy and bevacizumab, followed by maintenance durvalumab, bevacizumab and the PARPi olaparib in newly diagnosed advanced high grade OC after cytoreductive surgery (NCT03737643) [62]. If this trial demonstrates a good survival advantage, it may pave the way for immunotherapy to be included as part of routine OC treatment. Clinical trials using ICB alongside dual OC treatment combinations are summarised in Table 1.

#### 2.1.5. Other Adaptive Immunotherapeutic Strategies

Unfortunately, the success of ICB in the cancer patient population as a whole is limited, with only an estimated 13% of patients eligible for, and responding to, ICB therapy [63]. Alternative immunotherapeutic strategies employing the adaptive immune response include adoptive T-cell therapy (ACT). This approach involves the infusion of autologous or allogeneic antigen-specific T-cells that have been modified ex vivo into patients to stimulate a targeted immune response. Its success can be amplified by lymphodepletion prior to treatment, as demonstrated by complete tumour regression in 22% (20/93) of patients with metastatic melanoma. Impressively, in this study, 98% of participants remained in complete remission at 3 years [64]. Evidence of the cost-effectiveness of ACT compared to ipilimumab for second-line treatment in melanoma also makes it an attractive option [65]. In OC, a small study of 13 patients with no detectable lesion post primary surgery and cisplatin chemotherapy had a 3-year OS of 100% and PFS of 82.1% in the T-cell transfused group, versus 67.5% and 54.5%, respectively, in the cisplatin monotherapy group [66]. Nevertheless, ACT is limited by the technical challenge of generating sufficient numbers of CD8+ T-cells in vitro for infusion. Moreover, when ACT is administered into a high antigen burden environment, infused CD8+ T-cells often do not persist beyond a few days and thus cannot establish a sufficiently effective anti-tumour response to achieve a durable clinical effect. While this hurdle could conceivably be overcome through repeated ACT transfusions, this strategy is both technically demanding and costly. Moreover, the expression of co-inhibitory molecules such as T-cell CTLA-4 can lead to the inactivation of CD8+ TILs. As a result, the use of anti-CTLA-4 antibodies during initial autologous OC TIL culture could favour expansion of tumour-reactive CD8+ T-cells in TIL populations and provide a more robust anti-tumour response [67]. An interventional trial combined ipilimumab with ACT in six patients with advanced, recurrent, platinum-resistant HGSOC. This dual therapy resulted in tumour regression in all 6 patients (8–32% reduction in size of metastatic lesions), with 1 patient exhibiting stable disease for 12 months [68].

Other approaches are also aimed at maintaining CD8+ T-cell populations. Administering IL-2 has been shown to prevent the loss of CD8+ T-cells in animal models and clinical trials. More specifically, IL-2 promotes effector T-cell differentiation into memory cells and increases natural killer (NK) and CD8+ T-cell-mediated cytotoxicity [69]. However, a pilot study of progressive platinum-resistant OC demonstrated minimal anti-tumour effects with ACT following lymphodepletion and IL-2 therapy, with four out of six patients showing stable disease for only three months and two out of six patients for five months. This study concluded that the poor efficacy of ACT was associated with a high frequency of markers associated with effector T-cell dysfunction, including PD-1 and lymphocyte-activation gene 3 (LAG3) expressed on TILs, in addition to substantial PD-L1 expression in the tumour tissue from which the TILs were harvested [70]. LAG3 is another immune checkpoint protein that negatively regulates CD8+ and CD4+ T-cell activity and promotes Treg function. Persistent tumour antigen presentation induces chronic LAG3 expression on T-cells within the TME, leading to the impairment of CD8+ T-cell function through a progressive loss of cytokine production and cancer cell killing ability, a phenomenon dubbed T-cell functional ‘exhaustion’ [71]. Dual antibody blockade of LAG3 and PD-1 has been shown to significantly improve T-cell effector function and delay tumour growth in vivo in OC murine models [72]. The fact that PD-1 expression has been identified as a selective marker of tumour-reactive CD8+ TILs in OC highlights a subgroup of patients who may have a better response to ACT TIL therapy [73]. An interventional clinical trial will investigate the effect of combining T-cell therapy with the anti-PD-1 nivolumab, anti-CTLA-4 ipilimumab and LAG3 blocking antibody relatlimab in patients with metastatic OC (NCT04611126) [74].

The clinical application of ACT TIL therapy in OC is restricted by the proportion of patients with sufficient tumour-reactive CD8+ T-cell numbers. Around 80% of HGSOCs exhibit CD8+ T-cell TME infiltrates, but only 22% of these are characterised by a high CD8+ T-cell count. Similarly, while some 50% of mucinous and clear cell OCs display CD8+ T-cells within the TME, only 12% and 4%, respectively, have a high CD8+ T-cell counts [75]. In addition, the immunosuppressive TME environment can further hinder T-cell responses, limiting the clinical application of ACT. Efforts to enhance antitumour activity include chimaeric antigen receptor (CAR) T-cell therapy. CAR is an engineered hybrid combining an antibody variable region with a T-cell receptor (TCR) to stimulate immune cell responses targeted to a specific tumour antigen. CAR T-cell therapies have already secured FDA approval for the management of B-cell lymphoma following the success of clinical trials [6]. However, a combination of the financial outlay to support drug development, the need for close outpatient follow-up, the preventative management/treatment of complications (e.g., cytokine storm) and its modest success in treating solid malignancies continue to present obstacles to the technique’s further development and adoption [76]. Antigen targets for OC include mesothelin, mucin 16 (MUC16) and folate receptor (FR)-α. Mesothelin is a membrane glycoprotein that is overexpressed in OC (circa 55% of high/low-grade serous tumours), making it an attractive target for immunotherapy [77]. In this regard, the administration of anti-mesothelin CAR T-cells in six patients with recurrent OC stimulated an immune response with a clearance of pleural effusion demonstrated in one participant, suggesting a direct anti-tumour effect [78]. There were no adverse side effects reported and all patients had stable disease on imaging by one month. FR-α is even more commonly overexpressed in OC cells (>90% in non-mucinous OCs), where it modulates folate uptake to facilitate DNA synthesis and tumour cell proliferation [79]. In vivo, FR-α-specific CAR T-cells have demonstrated anti-tumour activity in OC mouse models [80], and comparable results have also been obtained in a phase I trial [81]. A summary of ongoing CAR T-cell clinical trials in OC is given in Table 2.

In order to enhance priming and activation of antigen-specific T cells, various cancer vaccination strategies have been developed, with 126 OC vaccine trials being registered on clinicaltrials.gov. A phase II trial of the FR-α-derived peptide vaccine for patients in remission from OC and endometrial cancer demonstrated a superior 2-year OS compared to controls (55% versus 40%, respectively) and doubling of 2-year PFS (90% versus 42.9%, respectively), underscoring the merit of directing the adaptive immune response to improve outcomes in OC [82]. To date, however, the vast majority of vaccine research has involved the use of DC precursors which function as part of the innate immune system, which will be covered next.

### 2.2. The Innate Immune Response

Adaptive immunity is triggered by antigen presentation to T-cells via innate cells such as macrophages or DCs. This aids pathogen clearance through antigen-specific T-cell functions and develops immunological memory. As such, these characteristics offer new avenues for therapeutic developments in OC given that they can be manipulated to amplify anti-tumour responses, improve the efficacy of immunotherapy and overcome the development of therapeutic resistance.

#### 2.2.1. Dendritic Cells

DCs are present in all tissues. Therein, they function as professional APCs to process and present tumour-associated antigens via MHC-I/II molecules and stimulate specific T-cell effector and memory cells. They thus provide a crucial link between innate and adaptive immune systems. OC cells secrete transforming growth factor (TGF)-β and prostaglandin (PG) E_2_ in the TME, leading to an increased expression of PD-L1 on DCs and inhibition of adequate CD8+ T-cells responses, thus driving immunosuppression [83]. High levels of DC infiltration are seen in the OC TME, but due to the downregulation of co-stimulatory molecules (e.g., CD80/86) and decreased expression of MHC-II, they are characterised by having a weak antigen presenting ability resulting in suppressed T-cell priming [84]. Moreover, the expansion and suppressive functions of Tregs are also dependent on inducible co-stimulator ligand (ICOS-L) stimulation by tumour plasmacytoid DCs and contribute to immunosuppression within the TME [85].

The APC function of DCs has been exploited in anti-cancer vaccine strategies, with the FDA’s approval of the first DC-based vaccine (DCV) for prostate cancer in 2010 [5]. A common strategy for DCV therapy involves harvesting peripheral blood monocytes differentiated ex vivo, which are then loaded with specific tumour peptides or whole tumour lysates prior to re-infusion [86]. A small trial of advanced OC patients in remission used DCV loaded with human epidermal growth factor receptor (HER) 2-derived peptide, which is overexpressed in OC and associated with poor prognosis [87]. No evidence of disease was observed in 6 of 11 patients by 36 months, and the 3-year OS was 90% [88]. Another DCV generated with antigenic FR-α peptides was administered to patients with advanced OC in remission following primary treatment. Patients who exhibited elevation of antigen-mediated cytotoxicity against FR-α had a significantly improved PFS of 39% over a median follow-up of 49 months. Importantly, no grade 3 toxicity was reported [89]. Whilst there was no comparative arm in this trial, PFS using this DCV was favourable compared to phase III trials in advanced OC incorporating bevacizumab with standard chemotherapy treatment for patients who had undergone debulking surgery (where appropriate): These women had a PFS of 27% at 4 years [90]. However, to put these figures into context, a study of olaparib treatment of platinum-sensitive relapsed *BRCA1/2* mutant OCs achieved a 48% PFS rate at 5 years [91]. Thus, while DCVs continue to be a promising immunotherapeutic strategy in OC, further improvements in treatment success are warranted before these can be considered for routine management.

Chemotherapy such as gemcitabine has been shown to enhance antigen presentation by inducing tumour cell apoptosis in vitro in colon carcinoma cell lines and holds the potential to enhance vaccine-related immunity [92]. An early clinical trial of platinum-sensitive relapsed OC demonstrated improved OS when combining DCV with gemcitabine and carboplatin compared to chemotherapy alone, with a median OS of 35.5 versus 22.1 months, respectively, as well as presenting a favourable safety profile [93]. Similarly, cyclophosphamide has been shown to augment DCV effects in OC due to its ability to deplete Treg numbers. Moreover, cyclophosphamide, bevacizumab and DCV triple therapy demonstrated a superior OS to bevacizumab and DCV or DCV alone in advanced, recurrent platinum-treated OC, with a 60% remission rate and a median PFS of 15 months amongst patients with a reactive T-cell vaccine response [94].

Other strategies to improve vaccine efficacy include the use of immunostimulatory cytokines. A preclinical melanoma mouse model study systematically evaluated 10 types of vaccines modified to incorporate immunomodulators. It demonstrated that granulocyte-macrophage colony stimulating factor (GM-CSF) provided the most durable and specific anti-tumour immunity compared to other immunomodulatory cytokines (e.g., IL-2, IL-4, IL-6 and IFN-γ) [95]. GM-CSF used in combination with IL-2 local adjuvant treatment in mouse models of lung cancer enhanced antigen presentation to increase CD8+ T-cell responses and induced immunological memory. This protected mice from subsequent tumour challenge, highlighting the potential of GM-CSF and DCV combination therapy to enhance the efficacy of anti-tumour responses [96]. In the clinical setting, superior survival outcomes were noted in advanced OC patients when a p53 peptide-based DCV was combined with subcutaneous GM-CSF compared to those without supplementary GM-CSF (median OS 40.8 versus 29.6, median PFS 8.7 versus 4.2 months, respectively) [97]. There are currently eight active trials investigating DCVs for OC (see Table 3). The success of DCVs in early trials has led to the first phase III trial of 678 patients with relapsed platinum-sensitive OC (HGSOC or endometroid histology) randomised to receive whole tumour lysate-loaded DCs in combination with bevacizumab, PARPi and chemotherapy agents (NCT 03905902) [98]. If clinical benefit is confirmed in this trial, the first DCV approved for OC may be anticipated.

#### 2.2.2. Natural Killer Cells

NK lymphocytes can detect and destroy virally infected or malignant cells without priming or prior sensitisation. They induce target cell apoptosis and produce anti-tumour cytokines such as IFN-γ (which is involved in priming the adaptive immune system by promoting maturation of DCs and inducing differentiation of CD4+ T-cells) [99]. NK function is governed by activating and inhibitory receptors that upon engagement with tumour cells can either induce or suppress their clearance. The immunosuppressive TME limits NK cell cytotoxicity by reducing the expression of activating receptors, such as NKp30. The expression of this receptor is substantially reduced on ascitic tumour-associated NK cells taken from HGSOC patients compared to their peripheral blood counterparts. Impaired NKp30 expression is associated with high levels of its ligand B7-H6 on the surface of OC cells, suggesting that defective expression is driven by chronic engagement of NK cells with B7-H6. This leads to NK cell hyporesponsiveness, characterised by impaired cytolytic activity and IFN-γ production [100]. Patient-derived NK treatment in vitro and in vivo in OC mouse models reduced tumour migration and invasion with the expansion of CD4+ and CD8+ T-cell populations, as well as IFN-γ production, leading to improved survival rates [101].

Inhibitory receptors impair NK cell cytokine secretion and cytotoxicity upon recognition of self-MHC-I ligands. While self-MHC-I molecules are downregulated in infected or malignant cells in order to avoid antigen-presentation to CD8+ T-cells, a key role of NK lymphocytes is to eliminate cells with low or absent MHC-I expression while maintaining tolerance to self-antigens. Cancer cells avoid this fate by overexpressing the non-classical MHC-I (mediator of inhibitory or activating NK cell stimuli) molecule human leukocyte antigen-E (HLA-E), which is a ligand of the inhibitory NKG2A receptor expressed by both NK and CD8+ T-cells. Chronic NK cell NKG2A expression therefore allows cancer cells to evade the anti-tumour immune response driven by these lymphocytes [102]. Early human trials of monalizumab, a NKG2A inhibitor, combined with an EGFR inhibitor in head and neck cancers previously treated with anti-PD-1/PD-L1 therapy showed a 31% ORR [103], leading to a phase III trial in this cohort (NCT04590963) [104]. Unfortunately, monalizumab monotherapy did not elicit any clinical effect in a study of 58 patients with recurrent gynaecological malignancies (HGSOC, squamous cervical and endometrial carcinoma) [105]. In mouse models of lymphoma, monalizumab and anti-PD-L1 therapy showed a 45% tumour growth control with expansion of CD8+ T-cells [103]. Regrettably, this combination had no comparable benefits in advanced HGSOC, with a 5.4% (2/32) ORR in a recent phase 1b clinical trial, prompting the consideration of alternative NK cell therapeutic approaches [106].

As previously discussed, there have been encouraging clinical results from adoptive T-cell therapy. However, a limitation of this treatment is the development of graft-versus-host disease (GVHD). NK cells provide a safer source of allogeneic cells for immunotherapy since their effector function is controlled by interaction with activating ligands on cellular targets, rather than by the expression of antigen-specific receptors on their surface, thus reducing the risk of GVHD. In this regard, adoptive NK cell transfer has demonstrated clinical effectiveness in early phase trials, notably by inducing remission in advanced myeloid leukaemia [107]. Tumour cells downregulate MHC-I molecule expression, which correlates with increased PD-1 expression on NK cells. This subsequently leads to defects in degranulation, cytotoxicity and INF-γ production [108]. This observation bore out in clinical trials, which demonstrated superior OS/PFS when combining pembrolizumab with allogeneic NK cells in advanced lung cancer patients with PD-L1 positive tumours, compared to pembrolizumab alone [109]. These findings lend support to the notion of using NK cell therapy to complement ICB for the treatment of solid tumours. However, although allogeneic NK cell therapy has shown clinical promise, there are no accessible ‘off the shelf’ NK cell products currently available. In the OC setting, intraperitoneal NK cell transfusions have the potential to bypass the depletion of transferred cells by the liver and spleen, thus maximising direct interaction with cancer cells. In this respect, early trials using intraperitoneal NK infusions, with or without preceding immunosuppressive conditioning [110] or combined with IL-2 and enoblituzumab (a monoclonal antibody that targets NK cell inhibitory ligand B7-H3) for OC patients are ongoing (NCT04630769) [111].

As intimated above, strategies to improve the anti-tumour NK cell therapy immune response include the addition of activating cytokines or their analogues. NK cells fall into two subsets: cytotoxic CD56^dim^CD16^+^ and immunoregulatory CD56^bright^CD16^−^ NK cells. The levels of CD56^bright^ NK cells are increased in OC ascites compared to those of peripheral blood [112]. Priming with the IL-15 receptor superagonist ALT-803 challenges this paradigm, as NK CD56^bright^ cells show markedly enhanced anti-tumour function, improving NK cell cytotoxicity, degranulation and cytokine production [113]. The development of this agent ensued from the fact that significant toxicity was reported in clinical trials using IL-15 (which promotes NK cytotoxic activity by its trans-presentation on DCs and macrophages via IL-15Rα). ALT-803 mimics trans-presented IL-15 and enhances NK cell activity against OC cell lines in vitro [114]. ALT-803 used in murine models of OC showed an NK-cell-dependent tumour reduction [115]. Related approaches have also been the focus of recent study. Human cytomegalovirus (CMV) infection can induce long-lived NK cells that exhibit enhanced specific cytotoxicity. NK cells manufactured from CMV positive haploidentical donors cultured with IL-15 were transfused to platinum-resistant OC patients following lympho-conditioning. A clinical effect was observed in three out of nine patients, with one patient experiencing a 48% tumour reduction [116]. Such encouraging results have led to a phase I trial testing the efficacy of using NK cells isolated from CMV positive donors and transfused to OC patients (NCT03213964) [117].

The success of engineering immune cells with CAR to improve tumour-specific killing has been demonstrated by CAR T-cell therapy in haematological malignancies and has led to a resurgent interest in developing CAR NK cell therapy. CAR NK cells can deplete cancer cells both in a CAR-dependent manner as well as by utilising NK cells’ natural cytotoxicity. NK cells can be isolated from peripheral blood or from induced pluripotent stem cells (iPSCs). CAR constructs are engineered with a stimulatory domain to improve cytotoxic NK cell killing and a tumour-associated target. One such domain is that of NKG2D, a stimulatory receptor expressed by NK cells [118]. The overexpression of the NKG2D ligand ULBP2 in tumour cells leads to chronic engagement and impaired function and is associated with poor prognosis in OC [119,120]. As such, CAR NKs targeting NKG2D ligands may rescue the cytotoxic NK cell response. In this regard, peripheral blood NK cells transfected with mRNA constructs encoding the chemokine receptor CXCR1 demonstrated enhanced NK cell infiltrate to the TME, with superior tumour control in mouse models of peritoneal carcinoma. When NKG2D CAR NK cell therapy was modified to include CXCR1 gene transfer, it demonstrated superior in vivo tumour control compared to NKG2D CAR NK cell alone [121].

The favourable clinical effect and safety profiles of targeting mesothelin in CAR T-cell therapy also identified it as an attractive option for CAR NK therapy. NK cells differentiated from human iPSCs were transfected with a mesothelin-directed CAR construct. This contained anti-mesothelin single chain variable fragments, the NKG2D transmembrane domain, the CD244 NK co-stimulatory domain and the CD3ζ signalling domain. OC mouse models treated with these NK cells demonstrated significantly reduced tumour growth and improved survival compared to those treated with non-CAR transfected NK cells. Notably, the in vivo activity of these CAR-modified NK cells was similar to that of CAR expressing T-cell therapy, but with significantly less toxicity [122]. Another preclinical study validated the use of CAR-engineered NK cells targeting mesothelin and demonstrated robust anti-tumour NK cell-dependent activity in OC mouse models [123]. The merit of this therapy is currently being evaluated in an early human trial for advanced OC (NCT03692637) [124].

#### 2.2.3. Macrophages

Monocyte-derived macrophages use pattern recognition receptors such as toll-like receptors (TLR) to identify pathogens. As phagocytes, they engulf these targets and present them to Th cells to engage the adaptive immune response, culminating in a vigorous anti-pathogen reaction. Circulating monocytes are recruited to the TME and differentiate to resemble classically activated M1 or alternatively activated M2 macrophages [125]. M1 phenotype polarization is driven in part by IFN-γ and exhibits anti-tumour properties characterised by a high capacity for antigen presentation. By contrast, M2 macrophages have poor antigen presenting ability, suppress CD4+ adaptive immunity, and promote the production of VEGF and immunosuppressive cytokines (IL-10 and TGF-β) [126,127]. Unsurprisingly, a high density of M2-like macrophages within the TME is associated with reduced PFS, [128] while a higher M1/M2 ratio correlates with improved 5-year survival in OC [129]. Macrophages also support tumour growth. Macrophage migration inhibitory factor (MIF) is a cytokine rapidly released by stimulated macrophages that promotes tumour cell migration, suppresses p53-mediated apoptosis and inhibits the anti-tumour immune response [130]. MIF is significantly overexpressed in OC (serous, mucinous and endometroid subtypes) compared to borderline or benign ovarian tissue and is implicated in the lack of NKG2D activation on NK cells [131]. Furthermore, MIF is associated with cancer invasiveness. Tumour-associated macrophages secrete matrix metalloproteases (MMPs) in an MIF-dependent manner, which promotes proliferation of tumour cells and TME neovascularisation. Moreover, MIF knockdown in OC cell lines has been shown to significantly inhibit invasiveness [132].

Preclinical work has focussed on strategies to reduce the number of immunosuppressive macrophages within the OC TME and/or suppress their function, such as by targeting the M2 macrophage pattern recognition receptor scavenger receptor A (SR-A). This receptor is upregulated in OC and promotes tumour cell invasion. Inhibiting SR-A in murine models of OC reduced tumour growth, revealing a potential therapeutic target for OC management [133]. Other studies have instead focussed on improving macrophage anti-tumour phagocytic activity. CD47 is a ligand for signal regulatory protein (SIRP) α, which is expressed on DCs and macrophages. The CD47/SIRPα axis suppresses the macrophage-mediated clearance of cancer cells. Importantly, CD47 is overexpressed in OC and is associated with adverse prognosis [134]. Anti-CD47 antibodies have been shown to stimulate macrophage tumour cell phagocytosis as well as antigen-specific CD8+ T-cell cytotoxicity, highlighting a potential synergy with ICB [135]. In this respect, CD47 antagonists and anti-PD-L1 reportedly elicit sustained anti-tumour effects and enhanced survival compared to anti-CD47 therapy alone in murine models of melanoma [136]. Moreover, anti-SIRPα antibody treatment increased NK cell and CD8+ T-cell infiltration and, when combined with anti-PD-1, yielded a more potent anti-tumour effect in mouse models of colon cancer [137]. A phase I/II trial is currently recruiting patients with advanced solid tumours for anti-CD47 or anti-PD-1 antibody and chemotherapy treatment (NCT04588324) [138].

CD24 interacts with the inhibitory receptor sialic acid binding Ig like lectin 10 (Siglec-10) on tumour-associated macrophages and promotes immune evasion by inhibiting phagocytosis. CD24 expression is also associated with enhanced cancer cell adhesion, invasion and metastasis [139]. It is commonly expressed in OCs (70–100% of cases), is rarely present in healthy tissue [140], and is an independent marker of poor prognosis in OC [141]. Anti-CD24 antibody therapy in both OC cell lines and murine OC models resulted in superior phagocytic clearance of tumour cells by macrophages, and correlating with reduced tumour growth in vivo [142]. Moreover, OC cell lines treated with anti-CD24-CAR NK cell therapy stimulated targeted cytotoxic activity against cancer cells, suggesting that anti-CD24 therapy may hold promise as a novel immunotherapeutic approach to be validated in future OC clinical trials [143].

Anexelekto (AXL) is a member of the Tyro3, AXL and Mer (TAM) receptor tyrosine kinase (RTK) family that orchestrates multiple processes, including cell survival, adhesion and migration. Tumour secretion of IL-10 promotes the upregulation and secretion of growth arrest specific 6 (GAS6; the ligand for AXL) in macrophages which, in turn, enhances cancer cell AXL signalling within the TME [144]. AXL is highly expressed in OC and intrinsically linked to epithelial–mesenchymal transition (EMT) and metastasis [145,146]. The inhibition of AXL in human OC cell lines suppressed tumour proliferation and increased sensitivity to cisplatin [147]. A phase Ib clinical trial comparing paclitaxel and AVB-S6-500 (which binds to GAS6 ligand, thus preventing AXL signalling) combination therapy with paclitaxel alone in platinum-resistant recurrent OC is ongoing (NCT03607955) [148].

TAM RTKs are also pleotropic inhibitors of both innate and adaptive immune responses. More specifically, they reduce the NK anti-tumour response, dampen innate cell-mediated inflammation, and regulate communication between dendritic and T-cells to control both the magnitude and quality of adaptive immune response [149]. AXL suppresses inflammatory TLR signalling in APCs by increasing suppressor of cytokine (SOCS)-1 and SOCS3 signalling [150]. Reversing TAM RTK-mediated immunosuppression may enhance ICB. Promising results from AXL-inhibitor and pembrolizumab combination therapy in AXL-positive advanced/metastatic lung cancer (NCT03184571) [151] led to its fast-track FDA approval in 2021. A phase I/II clinical trial of the GAS6-AXL signalling pathway blocker AVB-S6-500 in combination with anti-PD-L1 durvalumab in platinum-resistant, recurrent OC patients is ongoing (NCT04019288) [152] and, if successful, will justify phase III trials.

#### 2.2.4. Myeloid-Derived Suppressor Cells

MDSCs are a heterogeneous population comprised of myeloid progenitors and immature myeloid cells whose numbers expand during infection, inflammation and malignancy. MDSCs suppress the innate immunity by impairing NK activity [153] and decreasing macrophage production of IL-12, which is a major determinant of immune activation. The crosstalk between MDSCs and macrophages polarises M1 macrophages towards an immunosuppressive M2 phenotype [154]. These cells also secrete IL-10, which suppresses DC maturation [155]. MDSCs also produce arginase 1, which, in turn, reduces arginine availability to CD8+ T-cells—this amino acid (AA) is otherwise conditionally essential for their proliferation and for functional cytokine production [156]. MDSCs also inhibit CD8+ T-cell activity through the arginine-derived free radical nitric oxide (NO), which, when combined with the superoxide anion (O_2_^−^), produces peroxynitrites (PNTs). In turn, PNTs stimulate T-cell apoptosis, thus hindering adaptive immunity within the TME.

Compared to benign prostatic hyperplasia, prostate cancer displays increased MDSC numbers [157]. Their putative involvement in the pathophysiology of prostate cancer is supported by the observation that circulatory MDSC counts are significantly reduced following radical prostatectomy [158]. Analogously, when comparing the peripheral blood counts of patients with benign ovarian cysts, borderline or malignant tumours, MDSCs were noted to be independent predictors of ovarian malignancy, further suggesting that increased MDSC infiltration within the TME correlates with disease progression [159]. Importantly, melanoma studies have also reported that patients with high levels of circulating MDSCs have an inferior response to anti-PD-1 therapy, emphasising the potential of MDSC-targeted therapy to enhance the efficacy of ICB.

Selective MDSC depletion using an antibody targeted to the murine MDSC marker GR1 was shown to improve survival in a syngeneic OC mouse model [160]. However, no analogous human MDSC targets have yet been identified, focussing therapy towards the indirect suppression of MDSC function and reducing their recruitment to the TME instead. In this respect, breast cancer studies have shown that the chemotactic cytokine CCL2 drives MDSC migration to the tumour parenchyma [161]. Anti-CCL2 treatment has been shown to suppress intratumoural MSDC populations, inhibit both arginase 1 and NOS production and synergistically improve the effect of anti-PD-1 in murine models of lung cancer [162]. Tumour cells express ligands for the MDSC chemokine receptor CXCR2 which, upon binding, encourages MDSC infiltration into the TME. In this respect, anti-CXCR2 and anti-PD-1 combination therapy reportedly reduces tumour growth and improves the survival of mice inoculated with rhabdomyosarcoma compared to either agent alone. These mechanisms have been exploited in phase I/II trials combining anti-PD-1 with CXCR2 in melanoma (NCT03161431) [163] and with CXCR4 in metastatic pancreatic cancer (NCT02907099) [164] and could be a future approach for OC.

Other indirect strategies for manipulating MDSCs could further complement ICB. Liver-X receptor (LXR) is a nuclear hormone receptor that transcriptionally activates apolipoprotein E (*ApoE*). In melanoma, this has anti-tumour properties via LDL receptor related protein (LRP)1 receptor binding (which inhibits invasion) and endothelial LRP8 receptors (which suppress endothelial cell migration). In melanoma, the *ApoE* gene is positively regulated by the metastasis suppressor gene DNAJA4 and negatively regulated by metastasis-promoting miRNAs [165]. This allows melanoma cells to recruit endothelial cells and supports invasion. LRP8 receptors are also present on MDSCs and LXR agonism induces *ApoE*-mediated apoptosis of LRP8 positive MDSCs within the TME. Furthermore, LXR-mediated MDSC depletion with an LXR agonist was associated with a significant increase in IFN-γ and granzyme B positive TILs in vivo in mouse melanoma models, which enhanced the anti-tumour activity of anti-PD-1 therapy and significantly impaired tumour growth. This study also demonstrated the robust anti-tumour effects of LXR agonism in several in vivo murine cancer models, including OC. Consistent with observations in mouse models, the oral LXR agonist RGX-104 administered to patients with advanced solid malignancies or lymphoma in a phase I dose escalation trial reported a significant reduction in peripheral blood MDSCs. This also correlated with a substantial increase (216%) of peripheral tumour antigen specific CD8+ T-cells expressing glucocorticoid-induced TNFR-related protein (GIRT), a marker of T-cell activation [166]. Preclinical work on LXR agonist treatment led to validation of its use in a phase I trial, as a single agent or in combination with immunotherapy (nivolumab, ipilimumab or pembrolizumab) plus chemotherapy (carboplatin/pemetrexed) in advanced solid malignancies (NCT02922764) [167].

Since research developing innate immune system targets for OC management is in its infancy, there are no approved therapeutic applications based on targeting innate immune responses, either as monotherapy or in combination with other agents. The potential merit of combining innate and adaptive immune system targeting could maximise the efficacy of immunotherapy to enhance anti-tumour effects and generate durable results for OC patients.

## 3. Tumour Cell Intrinsic Pathways and the Immunosuppressive TME

There are several key oncogenic mutations in OC, including those affecting *T**P53*, *KRAS* and *PTEN*. These mutations influence immunoregulatory pathways within the TME which both promote cancer cell immune evasion and support tumour development. *TP53* gene mutations are almost universally observed in HGSOC [168], while *KRAS* and *PTEN* mutations occur in rarer OC subtypes. *KRAS* mutations are associated with low-grade serous OC (LGSOC) and mucinous tumours [169]. By contrast, functional *PTEN* loss is frequently detected in endometroid and clear cell OC morphological subtypes [170], and occurs only in around 7% of HGSOCs [168].

### 3.1. TP53

The transcription factor p53 plays an essential role as a tumour suppressor by regulating the cell cycle, DNA repair and apoptosis. CRISPR/Cas9-mediated *TP53* deletion in a murine syngeneic graft model of HGSOC led to increased MDSC infiltrates in both solid tumours and ascites via amplified expression of the chemoattractant CCL2, highlighting its role in contributing to TME immunosuppression [171]. Restoring p53 expression in vivo in murine hepatocellular carcinoma models resulted in tumour cell senescence and clearance as well as increased cytokine production and innate immune cell recruitment to the TME [172]. The miR-34 tumour suppressor microRNA (miR) family is transcriptionally upregulated by p53. miR34a/c are deregulated in cancer and control ULBP2 translation, leading to its downregulation and limiting activation of its target, the NKG2D stimulatory NK cell receptor [173]. PD-L1 expression is also downregulated by p53 via miR-34a binding to its 3′ untranslated region but results in an immunostimulatory effect instead in this instance [174]. Preclinical work using synthetic miR-34a mimics in human melanoma cell lines in vitro and in vivo in mouse models established stable native miR-34a expression and restored its tumour suppressor function, as demonstrated by apoptosis and reduced tumour cell growth in vitro and improved survival in vivo [175]. A clinical trial using miR-34 also demonstrated its delivery within the TME of solid tumours, including OC, with promising pharmacodynamic activity. Unfortunately, the trial was terminated due to fatalities relating to immunotoxicity, highlighting the difficulties associated with restoring p53-mediated tumour suppressor functions [176].

Mutant p53 is frequently overexpressed in OC cells, whereas wild-type p53 in healthy cells is generally found at low concentrations; this makes p53 an attractive target for immunotherapy [177]. Although the induction of antigen-specific immunity with synthetic p53 peptide vaccines has demonstrated T-cell responses in OC, these encouraging findings have failed to translate into improved survival [178,179]. This apparent discrepancy may be attributable to the presence of Tregs. TME Treg numbers are reduced by the chemotherapy agents cyclophosphamide [40] and gemcitabine, ref. [180] providing a rationale for combining them with p53 vaccines. Accordingly, a phase I trial of recurrent, platinum-resistant OC patients combing gemcitabine with a recombinant modified vaccinia ankara virus (MVA) expressing wild-type p53 (p53MVA) vaccine reported improved PFS in those with p53-reactive expanded CD8+ T-cells (the so-called ‘immunological responders’), compared to those who failed to elicit an immune response (PFS 6.7 versus 2.4 months, respectively) [181]. Combining cyclophosphamide with a p53-based vaccine in a phase II trial of recurrent OC successfully improved anti-tumour response, with 90% of patients demonstrating p53-specific T-cell responses and 20% exhibiting stable disease [182]. However, this response gradually diminished with subsequent vaccinations. A similar trend was reported in another trial using the p53MVA vaccine in patients with refractory colon and pancreatic cancers. Analysis of peripheral blood T-cell populations revealed that patients with higher frequency of PD-1 positive CD8+ T-cells had a significantly reduced induction of p53-specific CD8+ T-cell responses, suggesting that PD-1 blockade could enhance immune responses [183]. In this regard, a higher frequency of PD-1 positive CD8+ T-cells was found in peripheral blood samples of OC patients compared with those of healthy controls [181]. This provides a rationale to support further clinical studies of p53MVA ICB combinations in OC. Subsequent phase I clinical trials in advanced solid tumours (breast, pancreatic and hepatocellular carcinomas) combining pembrolizumab and p53MVA reported a clinical benefit in 3 of 11 patients, as demonstrated by a persistent p53-specific CD8+ T-cell response with subsequent vaccines. This resulted in an improved PFS of 30–49 weeks compared to no clinical benefit (10 weeks) in those who had borderline or undetectable p53 immune responses [184]. Another phase I trial is currently recruiting recurrent, platinum-resistant OC for treatment with p53MVA vaccine and pembrolizumab (NCT03113487) and may reveal a new treatment strategy for achieving durable anti-tumour responses in OC patients [185].

The functional relationship between the oncogenic Wilms’ tumour gene (*WT1*) and p53 is well established [186]. In OC, *WT1* expression is associated with prognostic indicators such as tumour grade and stage, and overexpression is consequently associated with poor OS [187]. Phase II trials of modified *WT1* peptide vaccines have demonstrated a 25–40% stable disease rate for gynaecological malignancies (including OC, cervical cancer, uterine sarcoma and endometrioid endometrial adenocarcinoma) [188,189]. A phase I trial further determining the efficacy of *WT1* vaccination as part of a nivolumab combination therapy in recurrent OC is ongoing (NCT02737787) [190]. Determining whether the efficacy of this intervention is matched in terms of safety profile will be key to determining the clinical viability of this approach.

While TGF-β has tumour-suppressive effects in early lesions, it becomes a key tumour promoter as cancers progress, promoting cell motility, invasion and immune evasion [191]. p53 is an essential contributor to TGF-β signalling. TGF-β type II receptors recruit, phosphorylate and activate Small Mothers Against Decapentaplegic (Smad) transcription factors [192], which promote EMT to support cancer cell survival and metastasis [193]. TGF-β is a potent immunosuppressor within the TME, as it promotes the expansion of Tregs, inhibits the generation and function of CD8+ T-cells and prevents the maturation of activated DCs [194]. This effect was observed in mouse models of colon cancer generated to be either TGF-β secreting or non-secreting. A DC-based vaccine in these models resulted in significantly lower, tumour-specific CD8+ T-cell responses, resulting in poorer survival in animals generated with a TGF-β secreting phenotype [195]. Moreover, using a nanoparticle-based delivery system to combine tumour antigen delivery to DCs with TGF-β small interfering RNA (siRNA) resulted in a 50% reduction of TGF-β within the TME. This was associated with the increased proliferation of CD8+ TILs and decreased Treg levels in the TME and featured a significant reduction in tumour growth compared to vaccine administered without TGF-β silencing in murine models of melanoma [196].

Furin is a proteolytic enzyme essential for activation of TGF-β1 and TGF-β2 pro-proteins. Vigil is an anti-tumour vaccine prepared from autologous tumour tissue, which is genetically modified to express short hairpin RNA to block furin, consequently downregulating TGF-β production. A phase II trial using Vigil in advanced OC in complete remission post cytoreductive surgery and chemotherapy reported a 27.6% 4-year PFS for the Vigil group versus 9.1% for the placebo control group [197]. These benefits have been replicated in other phase II OC trials, notably with a significant survival advantage noted in the subgroup of patients with wild-type *BRCA* status (see earlier) [198,199]. A phase I trial in advanced, recurrent OC (HGSOC, endometrioid and clear cell subtypes) also reported a survival advantage of *BRCA* wild-type OC when treated with Vigil and anti-PD-L1 atezolizumab combination therapy (NCT03073525) [200]. These promising results justify Vigil evaluation in later phase OC clinical trials.

### 3.2. KRAS and PTEN

The *KRAS* gene belongs to the family of *RAS* proto-oncogenes. RAS proteins are GTPases and operate a complex signalling network that regulates cellular functions such as proliferation, differentiation and apoptosis [201]. Murine models of lung carcinoma harbouring oncogenic *KRAS* mutations exhibited enhanced proliferation and immunosuppressive IL-6 production by myeloid cells within the TME [202]. The therapeutic inhibition of IL-6 in *KRAS*-mutated murine models of lung cancer reduced tumour progression and featured a decrease in tolerogenic macrophages, MDSCs and Tregs [203]. Post-transcriptional regulation of PD-L1 mRNA stability is induced by *KRAS* activation, causing an upregulation of PD-L1 expression on tumour cells [204]. Combining the *KRAS* G12C inhibitor AMG-510 and anti-PD-1 increased CD8+ TME infiltrates, causing marked tumour regression in vivo compared to either treatment alone [205]. A phase Ib trial recruiting over 1000 patients with *KRAS* mutated solid tumours to sotorasib monotherapy (a RAS GTPase inhibitor) or in combination with other treatments such as immunotherapy (pembrolizumab, atezolizumab), bevacizumab or chemotherapy (NCT04185883) [206] is ongoing and has potential clinical translational value for *KRAS* mutant OC (although this is typically confined to less common OC subtypes, such as LGSOC and mucinous OC).

Mutated *KRAS* is associated with phosphatidylinositol-3’-kinase (PI3K) signalling, which leads to the downstream activation of protein kinase b (AKT) and mammalian target of rapamycin (mTOR) [201]. This intracellular signalling pathway is responsible for regulating the cell cycle, metabolism and angiogenesis [207]. *PTEN* is an inhibitor of this pathway which limits cell proliferation when acting as a tumour suppressor. While *PTEN* loss in the fallopian tubal epithelium generates serous borderline tumours in mouse models [208], serous tubal intraepithelial carcinoma (STIC) remains the recognised precursor of HGSOC [209]. The PI3K/AKT/mTOR pathway is frequently deregulated in OC and is associated with an adverse outcome [210,211]. Activating PI3K mutations in mice-induced premalignant ovarian epithelial hyperplasia, when coupled with *PTEN* loss, led to the development of ovarian serous adenocarcinomas and granulosa cell tumours [212]. Both OC cell lines in vitro and OC mouse models showed enhanced anti-tumour effects and increased sensitivity to chemotherapy with PI3K and AKT inhibitors [213,214,215,216]. While approvals for PI3K inhibitors for the treatment of follicular lymphoma and mTOR inhibitors for renal cancer have been granted by the FDA, monotherapy targeting this pathway has provided underwhelming results in OC [217,218,219], such that no targeted therapies are currently available.

*PTEN* downregulates the expression of immunosuppressive cytokines and PD-L1 in a PI3K-dependent manner. As such, loss of *PTEN* control leads to the upregulation of PD-L1, as has been shown in advanced, triple-negative breast cancer cell lines [220]. PI3K inhibition combined with anti-PD-1 and anti-CTLA-4 significantly inhibited tumour growth and increased durable DC, NK cell and CD8+ T-cell responses compared to ICB alone in mouse models of this carcinoma subtype [221]. This preclinical research provides a rationale for combining PI3K inhibitors with ICB. In this respect, clinical trials are currently recruiting patients with metastatic or unresectable solid malignancies with *PTEN* or *PI3K* mutations for PI3K inhibitor (copanlisib) therapy alongside nivolumab or ipilimumab (NCT04317105) [222] or copanlisib and nivolumab for mismatch repair proficient colorectal cancer (NCT03711058) [223]. Therapeutic strategies targeting *KRAS* or PI3K/AKT/mTOR signalling pathways warrant further investigation, especially for rarer subtypes of OC such as LGSOC, which are notoriously chemoresistant.

The inactivation of tumour suppressor genes and/or activation of oncogenes also drives metabolic reprogramming of cancer cells. This plays an essential role in cancer progression to provide energy for rapid proliferation and survival (see section below). In particular, cancer cells’ high glycolytic activity needs to be met by markedly increased glucose uptake. Transmembrane glucose transport is the first rate-limiting step in glucose metabolism and is enabled by glucose transporters (GLUTs). In this regard, when acting as a tumour suppressor gene, *TP53* directly reduces *GLUT1* and *GLUT4* transcription and indirectly reduces that of *GLUT3* through the downregulation of NF-κB [224,225]. Furthermore, p53 transcriptionally activates parkin, which in turn promotes the degradation of hypoxia-inducible factor (HIF)-1α [226]. As HIF-1α activates the transcription of genes for both GLUTs and glycolytic enzymes [227], the resultant net effect is controlled glucose uptake. However, these metabolic controls are lost in mutated *TP53*, where metabolism is reorchestrated to support tumour growth. Loss of p53 functionality induces parkin deficiency. This subsequently leads to a downregulation of PTEN (a negative regulator of PI3K/AKT) [228]. The resulting activation of PI3K/AKT signalling promotes cancer cell glycolysis by directly regulating *GLUT1* and glycolytic enzyme expression [229]. Thus, aberrant p53 and *PTEN* loss drive the Warburg effect to support cancer cell glucose metabolism and growth. Specific mutations also alter amino acid provision. For example, *KRAS* mutant tumours upregulate the amino acid transporter SLC7A5 in order to meet the increased demand for protein synthesis allied with rapid cancer cell proliferation [230]. As such, genomic aberrations also in part contribute to TME metabolic reorchestration.

## 4. Metabolic Profile of the TME

Immune cell differentiation and functional activity hinges on their ability to efficiently metabolise available substrates. The metabolic demands of rapidly dividing tumour cells and impaired vascularisation deplete the TME of nutrients and oxygen, resulting in competition between cancer and stromal cells for limited resources. Tumour cell-driven metabolic reprogramming of the TME interferes with immune surveillance and promotes cancer progression (Figure 1). Targeting these metabolic perturbations could restore anti-tumour defences and overcome resistance to ICB therapy.

### 4.1. Glucose and Lactate

Malignancy-associated metabolic reprogramming features accelerated aerobic glycolysis; a phenomenon dubbed the Warburg effect. The Warburg effect supplies intermediates for macromolecule biosynthesis as well as nicotinamide adenine dinucleotide phosphate (NADPH) for reductive biosynthetic reactions/redox homeostasis to support rapid cell proliferation and resilience to environmental stressors such as hypoxia [231]. Increased aerobic glycolysis leads to lactate accumulation in the TME. In ovarian and pancreatic cancer cell cultures, lactate exposure stimulates IL-8 expression, which promotes tumour cell proliferation and migration [232,233]. Moreover, elevated serum lactate dehydrogenase (LDH; which converts pyruvate to lactate) correlates with shorter survival in OC [234] and is a predictor of platinum resistance [235]. By contrast, downregulating LDH in vitro inhibits proliferation in OC cell lines [234]. Analogously, reversing the Warburg effect, through the knockdown of pyruvate kinase (which regulates the rate-limiting final step in glycolysis) expression has been shown to greatly compromise tumourigenicity in lung carcinoma cell lines [236].

Increased glycolytic activity and the allied lactate build-up impacts both adaptive and innate immune cells. Indeed, lactate reduces NK cell activation and cytolytic function, increases MDSC numbers [237], inhibits DC differentiation [238] and promotes macrophage polarisation to a pro-tumourigenic M2 phenotype [239]. Under acidic conditions, CD8+ T-cell production of IFN-γ and IL-2 is impaired, and the expression of CTLA-4 is upregulated, rendering CD8+ T-cells sensitive to inhibitory signals. Low extracellular pH also inhibits the surface expression of CD25 and CD71, which are involved with CD8+ T-cell activation, collectively illustrating how a low pH environment contributes to local immunosuppression [240].

T-cell effector function is dependent on aerobic glycolysis, but effector T-cell metabolic exhaustion is mediated by both cell-extrinsic competition for glucose due to the rapid consumption of dividing tumour cells and cell-intrinsic mTORC1-driven metabolic reprogramming. Increased glyceraldehyde 3-phosphate dehydrogenase (GAPDH), a glycolytic enzyme, inversely correlates with IFN-γ production and is associated with increased PD-1 expression on CD4+ T-cells, promoting TME immunosuppression [241]. Intratumoural CD8+ T-cell populations with high PD-1 expression have significantly higher glucose uptake than those without in lung cancer patients [242], indicating that increased glycolysis may, in part, impart intrinsic immune resistance to tumour cells. This is further illustrated by the work of Cascone and colleagues [243], who showed that reduced T-cell infiltration and clinical resistance to adoptive T-cell therapy were associated with an upregulation of glycolysis-related genes in both lung cancers and melanomas which negatively reflect TIL levels. In addition, inhibiting glycolysis in these patient-derived cell lines enhanced T-cell mediated killing in vitro [243].

While aerobic glycolysis is important in the early stages of T-cell activation, it has become increasingly clear that the tricarboxylic acid (TCA) cycle/oxidative phosphorylation (OXPHOS) are critical to T-cell function. TILs are characterised by a decrease in mitochondrial mass compared to peripheral T-cells and consequently have a reduced respiratory capacity [244]. A contributing factor to this phenomenon is a progressive loss of peroxisome proliferator-activated receptor (PPAR)-γ coactivator 1-α (PGC-1α) on TILs (which programs mitochondrial biogenesis) driven by chronic AKT signalling from tumour-specific T-cells [244]. This reduces OXPHOS capacity, leading to metabolic exhaustion. A loss of mitochondrial mass and increased glucose uptake correlate with the upregulation of co-inhibitory receptors such as PD-1 and LAG3. Moreover, PD-1 represses the expression of the key metabolic regulator PGC-1α in CD8+ T-cells, further exacerbating their exhaustion [245]. In this regard, metabolic reprogramming of T-cells by enforced PGC-1α expression rescued metabolic function and induced superior anti-tumour responses with increased cytokine production, resulting in reduced tumour growth and increased survival of murine models of melanoma [244]. Similarly, a PGC-1α activator enhanced CD8+ T-cell activity and augmented PD-1 blockade efficacy, thereby reducing tumour volume in mouse models of colon cancer [246], suggesting a potential druggable target of TME metabolism that could improve immunosuppression and complement ICB therapy.

The anti-hyperglycaemic drug metformin inhibits respiratory-chain complex I, decreasing NADH oxidation and leading to a reduction in adenosine triphosphate (ATP) synthesis by OXPHOS [247]. Declining ATP levels lead to a rise in adenosine diphosphate (ADP):ATP and adenosine monophosphate (AMP):ATP ratios. In turn, this activates AMP-activated protein kinase (AMPK) pathways, which inhibit ATP-consuming anabolic reactions and favour ATP-generating catabolic processes to maintain energy stores [248]. Furthermore, AMPK inhibits the rate-limiting step of lipogenesis by phosphorylation of acetyl-CoA carboxylase, making it a key regulator of glucose and lipid metabolism [249]. Accordingly, the modulation of the AMPK pathway by metformin inhibits OC cell proliferation, both in vitro [250] and in vivo in murine models of OC [251]. Metformin treatment correlates with improved survival rates in diabetic patients with OC [252], and has the potential to enhance platinum sensitivity in OC stem cells ex vivo [253]. Head and neck squamous cell carcinoma (SCC) patients had greater CD8+ effector T-cell tumour infiltration when taking metformin, suggesting an immunomodulatory effect [254]. Both preclinical studies on melanoma, breast, lung and colorectal cancers [255,256] and a retrospective analysis of metastatic melanoma clinical trial data showed reduced tumour growth, enhanced CD8+ T-cell function and improved OS/PFS/ORR in patients treated with metformin and ICB compared to ICB alone [257]. This provides proof-of-principle that metabolic reprogramming can bolster TIL effector function to enhance immunotherapy [256]. However, in leukaemia models, metformin suppressed the cytotoxic activity of CAR T-cell therapy in vivo, ref. [258] suggesting that metformin may impede effector T-cell function is some settings. In OC, metformin has been reported to increase IFN-γ, perforin and granzyme B production by CD8+ T-cells, in both in vitro and in vivo murine models. This was also associated with decreased MDSC activity due to the metformin-mediated suppression of HIF-1α, which is critical for the induction of CD39/CD73 ectoenzyme activity on MDSC subsets [259]. CD39/CD73 produce extracellular adenosine which accumulates in the TME and inhibits anti-tumour T-cell responses by reducing antigen-induced proliferation and IL-2 and IFN-γ production [260]. Early phase clinical trials are thus exploring metformin and anti-PD-1 combination therapy in advanced non-small-cell lung cancer (NCT03048500) [261], colorectal cancer (NCT03800602) [262] and metastatic head and neck cancer (NCT04414540, NCT03618654) [263,264] and may suggest whether this approach has a future role in OC management.

### 4.2. Lipids

In response to glucose starvation, fatty acid (FA) oxidation (FAO) and lipid metabolism more broadly is also utilised by tumours to fulfil their energetic requirements. Tregs rely on intrinsic FAO for differentiation as demonstrated in mouse models of colon cancer [265]. FAO also promotes macrophage-mediated tumour cell migration in vitro in hepatocellular carcinoma cell lines [266]. CD8+ T-cells starved of both glucose and oxygen showed both an increased triglyceride turnover and mitochondrial catabolism of FAs [267]. This suggests that metabolically stressed CD8+ T-cells are dependent on FA catabolism and that overcoming TME immunosuppression is reliant on FAs. In this regard, the PPAR transcription factor family also upregulate genes involved in FA transport and FAO [268]. Thiazolidinediones are potent binders of PPAR-γ and are used to treat diabetes as they promote adipogenesis and FA uptake. Treatment with PPAR/PGC-1α complex agonists in mouse models of melanoma [267] and lung cancer [269] promoted FA catabolism, correlating with improved CD8+ T-cell effector function and increased acyl-CoA dehydrogenase long chain (LCAD) expression, an enzyme involved in FAO. When this agonist was combined with PD-1 blockade, it yielded greater CD8+ IFN-γ positive T-cells than PD-1 blockade alone and synergistically reduced tumour growth in vivo in lung cancer murine models [269].

FA catabolism is required for CD8+ T-cell activation and is essential for the proliferation and generation of T-cell memory [270]. 4-1BB (CD137) is an activation-induced co-stimulatory molecule found on the surface of T-cells, NK cells and DCs. Agonistic 4-1BB therapy in vitro has been shown to increase glucose metabolism to enhance CD8+ T-cell proliferation. It also induces liver kinase (LK)B1, a primary upstream kinase of AMPK pathways, leading to phosphorylation of acetyl-CoA carboxylase (ACC) to enhance FA metabolism [271]. Combining PD-1 blockade and anti-4-1BB antibody resulted in reduced tumour growth and significantly increased CD8+ T-cells and NK cells within the TME in mouse models of lung cancer [272]. Similarly, melanoma ACT studies showed that 4-1BB co-stimulation may also improve TIL survival and cytolytic activity in vivo in mouse models [273]. A phase I/Ib study will review the clinical effects of IL-2-primed ACT of autologous CD8+ T-cells in combination with the anti-4-1BB antibody utomilumab in platinum-resistant OC patients (NCT03318900) [274]. The multicentre phase II JAVELIN Medley trial (NCT02554812) [275] of 620 patients with advanced stage solid tumours is also evaluating the effect of PD-1 blockade alongside anti-4-1BB antibody and OX40 agonist (see section below on hypoxia) to assess the potential of metabolic targeting to enhance ICB therapy efficacy.

Fatty acid synthase (FASN) is a key lipogenic enzyme. The overexpression of FASN is associated with poor survival in melanoma [276], breast [277], pancreatic [278] and OC [279,280]. Data from OC murine models indicate that FASN inhibitors can restore sensitivity to cisplatin and tumour proliferation [281]. DCs isolated from OC ascites are characterised by high FASN expression which correlates with defective antigen presenting abilities, resulting in inactivation of anti-tumour T-cells. In particular, high FASN expression correlated with lower intratumoural effector CD8+ memory T-cells. Treatment with the FASN inhibitor cerulenin correlated with increased CD8+ IFN-γ positive T-cell numbers, with enhanced granzyme B release within the OC TME. Moreover, cerulenin reduced the lipid content of tumour-infiltrating DCs, which correlated with an enhanced capacity to induce CD8+ T-cell proliferation in murine models of OC [282]. In this respect, the first human study of FASN inhibitor (TVB-2640) monotherapy in advanced tumours showed a disease control rate of 42% (29/69) compared to 70% (37/53) in combination with paclitaxel and exhibited an 11.8% partial response rate in the OC group. Importantly, it had a favourable tolerability profile [283]. FASN inhibition could thus provide a mechanism to restore anti-tumour immunity.

OC cell lines exhibit upregulation of the sterol regulatory element-binding protein gene (SREBF)-1. This is associated with increased expression of lipid synthesis genes and downregulation of SREBF-2, which suppresses regulation of cholesterogenic target genes [284]. Excess lipids in cancer cells are converted into triglycerides and free cholesterol is esterified to cholesteryl esters by acyl-CoA cholesterol acyltransferase 1 (ACAT1). In this regard, ACAT1 knockdown OC cell lines exhibit significantly reduced cell proliferation and migration, as well as an increased sensitivity to cisplatin [285]. Inhibiting ACAT1 activity enhanced proliferation of CD8+ T-cells and, when combined with anti-PD-1, displayed superior control of tumour progression and survival in mouse models of melanoma [286]. However, targeting cholesterol in the TME has yet to be explored in OC.

The mTOR signalling pathway is also involved in lipid metabolism. Herein, it controls FA uptake through activation of transcription factor sterol receptor element binding protein-1 (SREBP-1) on CD4+ T-cells, which promotes FA synthesis. mTOR signalling also induces PPAR-γ in adipocytes to facilitate FA uptake [287]. Studies of metastatic renal cell carcinoma patients using the mTOR signalling inhibitor everolimus reported increased CD8+ T-cell infiltrates and changes to Treg/anti-tumour Th1 balance, thus potentially improving responses to immunotherapies [288]. Such benefits have been demonstrated in murine models of oral SCC, which had improved survival with rapamycin-mediated mTOR inhibition combined with anti-PD-L1 treatment compared to either treatment alone [289]. Investigations have been extended to anti-CTLA-4 therapy which, when combined with rapamycin and an anti-cancer vaccine in vivo in mice, resulted in CD8+ T-cell expansion, IFN-γ production and differentiation towards a memory phenotype as well as featuring enhanced FA metabolism and an increased respiratory capacity [290]. An early phase trial of sirolimus (another mTOR inhibitor) with the anti-PD-L1 antibody durvalumab in lung cancer patients is ongoing (NCT04348292) [291]. This will establish the merit of mTOR inhibitors in cancer immunotherapy that may ultimately be translated in OC.

### 4.3. Hypoxia Is a Key Modulator of the TME

As solid tumours develop, large areas become deprived of oxygen and nutrients due to increased cancer cell metabolism and disorganised vasculature. Tumours compensate for hypoxia by promoting angiogenesis to support their growth. Endothelial cells coordinate vessel expansion under the control of Notch signalling, which arrests angiogenic proliferation [292]. *PTEN* is required for Notch function, and as such, *PTEN* deletion leads to loss of Notch-mediated control, resulting in constant pro-angiogenic signalling [293]. Many tumours, including OC [294], also overexpress VEGF within the TME, which stimulates neovascularisation and increases vascular permeability. This chaotic formation of pathological vessels maintains a hypoxic state within the TME.

Moreover, VEGFs also promote immune evasion by recruiting Tregs, MDSCs and impairing DC activation and differentiation (Figure 2) [295]. Anti-VEGF treatment of mouse models of melanoma significantly increased CD8+ T-cell numbers as well as their IFN-γ production, granzyme B release and perforin gene expression within the TME [296]. This improvement of effector TIL function provides a rationale for investigating the potential synergistic anti-tumour effect of anti-VEGF treatment with immunotherapy. As such, an early-stage clinical trial of OC patients with p53-mutated HGSOC or endometroid carcinoma is investigating the effect of the anti-VEGF bevacizumab combined with the anti-PD-L1 antibody atezolizumab (NCT04510584) [297]. Anti-VEGF treatment in mouse models of colon cancer and melanoma associated with the upregulation of the co-stimulatory T-cell receptor OX40 [296]. OX40 has an important role in both CD4+ and CD8+ T-cell activation, expansion and survival, as well as regulating CD8+ T-cell memory capacity, making it an appealing target for combination with immunotherapy [298]. Anti-PD-1 combined with an agonistic OX40 antibody in OC murine models increased CD8+ T-cell function and markedly reduced tumour growth, such that 60% of animals were tumour-free at 90 days [299]. In this regard, a phase II trial of relapsed OC managed with anti-PD-1 or anti-CTLA-4 antibody with OX40 agonist combination therapy is ongoing (NCT03267589) [300].

The most widely recognised pathway enabling tumour cell survival in the hypoxic TME is the HIF pathway. The HIF-1α subunit is stabilised under hypoxia and binds to HIF-1β to initiate the transcription of over 100 target genes [301], including VEGFs (to promote tumoural neovascularisation) and *GLUT1* (to maintain glycolytic substrate provision). The HIF pathway also promotes EMT, encouraging cancer cell growth and metastasis [298]. Higher HIF-1α expression is observed in OC tissues and metastatic lesions than in benign fallopian tubes, suggesting that HIF-1α plays a role in OC progression [302]. HIF-1α is associated with worse survival outcomes in OC [303], and reduced expression is correlated with greater cisplatin sensitivity [304,305]. HIF-1α fosters an immunosuppressive TME by promoting Treg, MDSC and macrophage recruitment [306]. Hypoxia regulates the expression of inhibitory checkpoint proteins. As previously discussed, HIF-1α- mediates adenosine accumulation within the TME (due to upregulation of enzymes CD39 and CD73). This activates the adenosine receptor A2aR on tumour cells and CD8+ T-cells which subsequently upregulates their PD-1 and CTLA-4 expression [307]. CD8+ T-cells also upregulate LAG3, TIM3 and PD-1 expression in response to hypoxia in a HIF-1α-dependent manner [308,309]. Moreover, the upregulation of PD-L1 on tumour cells, MDSCs, Tregs, M2 macrophages and DCs is HIF-1α-dependent [310]. In this regard, PD-L1 blockade under hypoxic conditions induced CD8+ T-cell activation and was accompanied by the downregulation of IL-6 and IL-10 and reduced MDSC infiltrates in murine models of melanoma, colon, lung and breast carcinoma [311]. This suggests that synchronised PD-L1 blockade with HIF-1α inhibition could overcome ICB resistance and represent a novel approach for OC immunotherapy.

Topoisomerase I inhibitors such as camptothecin have also been shown to inhibit the HIF pathway. A phase II trial using the enhanced delivery camptothecin nanopharmaceutical CRLX101 combined with bevacizumab in 19 patients with platinum-resistant relapsed OC achieved durable partial responses in 3 patients (16%), tumour reduction in 14 (74%) and was well tolerated [312]. Moreover, this agent was well tolerated in combination with bevacizumab, resulting in an 18% ORR in phase II trials in patients with recurrent OC [313]. Decursin is a pyranocoumarin which has been shown to inhibit HIF-1α accumulation in human cancer cell lines grown under hypoxic conditions. When applied in vivo in mouse models of lung and colon cancer, it enhanced CD4+ and CD8+ T-cell expansion and attenuated both PD-1 expression and Treg accumulation, highlighting its potential synergy with ICB therapy [314]. Preclinical work on targeting the HIF pathway and anti-VEGF to enhance the efficacy of immunotherapy has led to a phase III trial currently recruiting 1,431 advanced renal cell carcinoma patients to evaluate pembrolizumab (anti-PD-1 antibody) and lenvatinib (VEGF receptor R1 (VEGFR1), VEGFR2 and VEGFR3 kinase inhibitor) with or without belzutifan (selective inhibitor of HIF-2α) versus quavonlimab (anti-CTLA-4 antibody) and lenvatinib (NCT04736706) [315]. If this trial successfully demonstrates good anti-tumour effects with low toxicity, this strategy could be replicated for future OC studies.

### 4.4. Amino Acids Metabolism and Immune Suppression in the TME

#### 4.4.1. Glutamine

AA metabolism supports protein biosynthesis, the production of nitrogen-containing metabolites for nucleic acid synthesis and intermediates for the TCA, as well as maintains intracellular redox balance. Moreover, it supports the proliferation and survival of cancer cells under nutritional or oxidative stress. In this regard, glutamine is an essential AA for tumour survival [316]. Mitochondrial glutaminase catalyses the hydrolysis of glutamine to glutamate, which can be converted to α-ketoglutarate (α-KG) for the TCA cycle [317], or used to support citrate production to generate cytoplasmic acetyl-CoA, which is a precursor for de novo fatty acid synthesis [318]. Indeed, reductive glutamine metabolism is a major carbon source for fatty acid synthesis under hypoxic conditions or when mitochondrial respiration is impaired [319]. Glutamine metabolism is controlled by both tumour suppressor genes and oncogenes. Mitochondrial glutaminase, encoded by the *GLS2* gene, can be transcriptionally induced by p53 [320]. In *KRAS*-mutated pancreatic cancer, the reprogramming of glutamine metabolism increased the NADPH/NADP+ ratio to maintain a stable redox state [321]. The overexpression of glutaminase in cancers helps to fulfil metabolic demand and confers platinum resistance in OC [322]. Inhibiting glutaminase has been shown to restrict tumour growth in both OC and other cancers [323,324,325], as well as sensitizing OC cells to paclitaxel chemotherapy [326,327]. Taken together, the exogenous manipulation of the glutamine metabolism within the TME may impact on the metabolic support of tumour growth.

AA metabolism also impacts tumour immunity. Blocking glutamine metabolism using the glutamine antagonist JHU08 in melanoma mouse models demonstrated how activated T-cells adapt to TME glutamine depletion by markedly upregulating OXPHOS and utilising acetate as a carbon source in the TCA cycle. This is reflected by an increase in acetyl-CoA levels in glutamine-restricted T-cells. These metabolic alterations allow T-cells to maintain their cellular AMP:ATP ratio and physiological function. By contrast, cancer cells are unable to compensate energetically when treated with a glutamine antagonist. This leads to increased AMP:ATP ratios, the activation of AMPK and the downregulation of the oncogene transcription factor c-MYC. Since AMPK and c-MYC are both crucial regulators of glucose influx [328,329], cancer cells’ ability to maintain Warburg metabolism is thus markedly reduced. Accordingly, JHU083 treatment in vivo in mouse models of melanoma combined with either ACT or anti-PD-1 antibody exhibited significant expansion of TIL numbers, resulting in improved tumour control and prolonged survival when compared to either agent alone [330]. Furthermore, targeting glutamine metabolism using the glutamine antagonist 6-diazo-5-oxo-L-norleucine (DON) inhibits TME infiltration by MDSCs and promotes the reprogramming of tumour-associated macrophages to the pro-inflammatory M1 phenotype in mouse models of breast cancer [331]. Intervention can also be achieved at the enzymatic level. In this vein, the glutaminase inhibitor CB-83 has been shown to synergise with anti-PD-1 treatment in mouse models of clear cell OC [332]. Collectively, these examples suggest that glutamine metabolic blockade could condition the TME to enhance its response to immunotherapy. Unfortunately, the use of DON in human trials was abandoned due to gastrointestinal toxicity. Constructing a prodrug to improve DON therapeutic index or testing low-dose regimes are potential strategies for redevelopment given the potential benefits of targeting TME glutamine metabolism as a complementary intervention for ICB or ACT immunotherapy.

#### 4.4.2. Arginine

While tumours rely on extracellular arginine to support their metabolic requirements, it is also important for T-cell activation. Since arginine is crucial to support both cancer cell proliferation and immune responses, it has been explored as a potential anti-cancer therapy. Arginine is used as a substrate for nitric oxide (NO) synthase (NOS), which oxidises it to NO and citrulline. NO production by macrophages can stimulate macrophage activity and CD8+ T-cell-mediated cytotoxicity. However, arginine is also a substrate for arginase 1 (ARG1), which hydrolyses it into ornithine. In turn, this is used to generate polyamines, which support cancer cell proliferation. The contrasting potential anti-tumour effects of arginine metabolism are thus dependent on the balance of NOS and ARG1 activity [333,334]. The overexpression of ARG1 from M2 macrophages and Tregs within the TME enhances arginine degradation to ornithine, which subsequently promotes cancer cell growth, restricts NOS-mediated cytotoxicity and limits TME arginine availability for T-cells, thereby suppressing their activation [335]. Accordingly, early results of the ongoing clinical trial (NCT02903914) [336] using ARG1 inhibitor monotherapy or combined with anti-PD-1 for advanced solid tumours has shown good tolerability with ORR and disease control of 3%/28% and 6%/37%, respectively [337]. Furthermore, melanoma mouse model studies focussed on L-arginine supplementation showed that this intervention improved the survival of antigen-activated T-cells [338]. Moreover, L-arginine supplementation in combination with an anti-PD-L1 antibody amplified CD8+ T-cell infiltration and activity, resulting in significantly prolonged survival of mouse models of osteosarcoma [339].

OC cells are auxotrophic for arginine and depletion of this AA causes OC cell cycle arrest and leads to cell death by autophagy [340]. Under arginine restriction, this AA can be recycled from citrulline by argininosuccinate synthase (ASS). In contrast to T-cells, which can upregulate citrulline import and ASS in response to arginine depletion, OC cells do not express ASS. The result is that while T-cell proliferation can continue under arginine restriction, cancer cells remain reliant on extracellular sources [341]. In this regard, arginine depletion with PEGylated arginine deiminase (ADI-PEG20) in advanced melanoma patients resulted in stable disease in 38% (9/24) of patients and, when combined with pembrolizumab in advanced solid tumours, had a 24% (6/24) ORR, which was associated with increased TILs [342]. This combination therapy had a 40% rate of neutropenia but was otherwise well tolerated. Early results from a phase Ib trial using ADI-PEG20 in synergy with the anti-PD-1 pembrolizumab in advanced stage solid tumours demonstrated increased T-cell TME infiltrates (83%) and a 24% partial response rate, leading to this being extended to phase II trials [343]. ADI-PEG20 treatment in mouse models of small-cell OC inhibited tumour growth [344]. As this rare OC subtype has been reported to respond to anti-PD-L1 treatment, this raises the possibility of considering combination therapy for these patients [345].

#### 4.4.3. Tryptophan

The essential amino acid tryptophan is catabolised by the rate-limiting enzymes indolamine-2,3-dioxygenase (IDO) and tryptophan-2,3-dioxgenase (TDO), resulting in tryptophan depletion and the production of kynurenine metabolites [346]. Kynurenine is either fed into the TCA cycle or converted to kynurenic acid. The fact that tryptophan metabolism plays a crucial role in immune tolerance was first identified at the foeto-maternal interface, where IDO expression prevents T-cell-mediated foetal rejection [347]. This concept is rooted in the fact that tryptophan depletion suppresses effector T-cell function [348]. Macrophages and DCs express IDO [349], which locally depletes tryptophan and arrests cytotoxic T-cell proliferation [350]. High levels of IDO in OC are correlated with poor survival outcomes [351] and preclinical work has pointed to a role for IDO in TME immunosuppression. Mouse OC models with IDO overexpressing cells displayed a significant increase of peritoneal disease and ascites compared to controls, suggesting that IDO is linked to peritoneal dissemination. This phenomenon was accompanied by reduced numbers of CD8+ T-cells and NK cells, as well as increased TGF-β levels in ascitic fluid. Treatment of these models with the IDO inhibitor 1-methyl-tryptophan suppressed tumour dissemination and reduced ascitic TGF-β concentrations [352]. Moreover, a mouse model of OC established with an IDO downregulated cell line increased NK cell infiltration to the TME and reduced tumour growth [353]. Similarly, IDO inhibition in mouse models of metastatic liver, bladder and lung cancers caused an increased TIL proliferation and reduced Treg infiltrate in the TME, which were associated with delayed tumour growth [354,355,356]. IDO expression on TILs has also been found to be higher in an anti-PD-1 resistant model of lung cancer compared to ICB-sensitive controls, suggesting that its blockade could overcome ICB failure [357]. Although initial clinical IDO inhibition data was promising for use in conjunction with ICB [358,359], the first phase III trial of metastatic melanoma patients failed to demonstrate a survival advantage of anti-PD-1 and the IDO inhibitor epacadostat combination therapy over anti-PD-1 monotherapy [360]. However, a murine breast cancer model treated with a DCV containing IDO-silenced DCs demonstrated enhanced CD8+ T-cell activity, reduced Treg infiltrates and decreased tumour size compared to antigen-loaded DC-based vaccine without concomitant IDO silencing [361]. These findings suggest that IDO inhibition may have a potential role in improving anti-tumour vaccine efficacy.

The lack of translational clinical effect with IDO inhibitors inspired recent work on the transcription factor aryl hydrocarbon receptor (AhR), whose high cytoplasmic expression in OC correlates with poor prognosis [362]. The tryptophan metabolites, kynurenine and kynurenic acid are both AhR receptor ligands. Kynurenine-AhR-dependent signalling is implicated in TME immunosuppression by promoting Treg differentiation [363] and driving the recruitment of immunosuppressive tumour-associated macrophages to the TME [364]. In the same vein, enzyme-mediated depletion of kynurenine has been shown to work synergistically with anti-CLTA-4 to reduce tumour volume and improve survival in murine models of breast cancer [365]. In addition, the selective blockade of AhR reduced the tumour growth of mouse models of IDO-expressing melanoma. This was associated with repolarisation of tumour-associated macrophages to an M1 phenotype and increased antigen-specific T-cell priming, demonstrating this approach’s potential in reversing TME immunosuppression. In this regard, AhR inhibition in combination with anti-PD-1 antibody significantly reduced tumour growth and improved survival compared to either treatment alone in these models [366]. The first human trial to review the safety profile and dose regimen of an AhR inhibitor, BAY 2416964, is currently recruiting patients with advanced lung, hepatic, colorectal and urothelial cancers (NCT04069026) [367]. There are another two ongoing clinical trials using AhR inhibitors with immunotherapy for the treatment of advanced solid tumours, one combining BAY 2416964 with pembrolizumab in hepatic, lung and urothelial carcinomas (NCT04999202) [368] and the other combining the AhR inhibitor IK-175 with nivolumab (NCT04200963) [369]. Although there have been no trials of AhR inhibition specifically targeting OC to date, the putative role of AhR in the disease warrants their consideration.

L-amino acid oxidase IL-4-induced-1 (IL4I1) is an enzyme found to be associated with AhR induced gene expression more than other tryptophan catabolic enzymes in human tumours [370]. IL4I1-mediated tryptophan metabolism yields AhR receptor ligands, including kynurenic acid and indoles. IL4I1 is overexpressed in OCs compared to benign ovarian tissues, indicating a likely role in carcinogenesis and/or metastasis. A large study of 32 tumour samples from the cancer genome atlas (including OC) comparing high versus low-IL4I1 expressing tumours demonstrated increased numbers of immunosuppressive cells (e.g., Tregs and MDSCs) in the TME and suppressed proliferation of CD4+ and CD8+ T-cells in high IL4I1 expressing lesions [370]. Studies were extended to mouse models of B-cell lymphoma generated with IL4I1 deficient B-cells. These models demonstrated a greater number of classical DCs within the TME compared to controls, suggesting a greater antigen-presenting capacity, in addition to higher CD8+ T-cell expression of killer cell lectin-like receptor (KLR)G1, a marker of activated CD8+ memory T-cells [371]. This has been replicated in vitro using a B-cell lymphoma model. In this context, tolerogenic DCs exhibited increased IL4I1 expression, which was associated with reduced CD8+ T-cell proliferation [372]. Furthermore, leukaemic cell lines overexpressing IL4I1 promote the M2 polarisation of tumour-associated macrophages in vitro [373]. The immunosuppressive effects of IL4I1 could be used to enhance ICB response. When studying IL4I1 levels in ICB-treated advanced melanoma patients, it was noted that nivolumab therapy correlated with the induction of both IL4I1 and IDO1 and resulted in AhR activation. Notably, IL4I1 was significantly elevated in ipilimumab-naïve patients who developed progressive disease, suggesting that IL4I1 is involved in ICB resistance [370]. A retrospective analysis of IL4I1 expression in melanoma patients will determine its predictive value in response to immunotherapy (NCT04253080) [374]. IDO-targeted drugs do not similarly inhibit IL4I1; this implicates IL4I1 expression as a resistance mechanism against IDO inhibitors given that this maintains AhR ligand production. This may explain their failure to improve the response to ICB in phase III trials. Overall, IL4I1 is implicated in diminishing anti-tumour CD8+ T-cell responses, thereby promoting resistance to ICB. Consequently, it is also associated with reduced patient survival. IL4I1 blockade is therefore a promising target with the potential to supplement immunotherapy, although it has yet to be considered as a potential metabolic target in the context of OC.

## 5. Concluding Remarks and the Future of Immunotherapy in OC

Despite the success of ICB in certain malignancies, both as monotherapy and combination therapy, it has only yielded modest benefits in the OC setting and not without significant toxicity. *BRCA* mutated OCs feature greater TIL levels (and mutational burden), which clinically translate in improved survival. This makes this subset of tumours a tantalising case for future treatment with ICB [50,51]. VEGF promotes intratumoural Treg recruitment and ineffectively attempts to resolve TME hypoxia, justifying pairing anti-VEGF targets with ICB. This strategy has shown promising survival outcomes in early phase trials [51]. Whether platinum-based chemoimmunotherapy regimes truly benefit from the addition of standard OC treatments such as anti-VEGF or PARPi remains unanswered but will be addressed by the ongoing phase III clinical trials discussed herein [54,62,375,376].

Overcoming cancer cells’ ability to evade and suppress antigen-mediated T-cell activity within the TME is crucial for the success of immunotherapy. Engineered CAR T or CAR NK-based therapies can enhance both specificity and affinity for tumour antigens, eliciting a stronger anti-tumour immune response [6]. However, the clinical utility of CAR T-cell therapies in OC is likely limited by the modest fraction of patients displaying reactive TILs (4–22% in OC) [75] combined with the impact of the systemic depletion of transferred T-cells by the liver and spleen. In this respect, it is hoped that a phase I trial determining the efficacy of intraperitoneal NK cell therapy in OC may result in more reliable immune cell delivery [111]. The robust antigen-presenting ability of DCs has also been co-opted for developing immunotherapy. The ability of DCVs to stimulate the crucial link between innate and adaptive immune systems has translated into improved survival in early OC trials [89,93,94,97], leading to the first phase III trial of the autologous DCVAC/OvCa vaccine in platinum-sensitive recurrent OC. If improved survival is observed, we may anticipate the first innate immune cell-based immunotherapy to be approved for OC [98].

Immunotherapies must be tailored to molecular tumour subtypes and individual TME immunogenic landscapes. Patient selection on the basis of biomarkers (PD-1/PD-L1/IL4I1 expression, TIL/Treg ratio) and mutational profiles (MSI, *BRCA* status, *p53/KRAS/PTEN* mutation) that correlate with improved immunotherapy response will help identify those patients most likely to achieve durable benefit. Mutant p53 is almost universally expressed in HGSOC and is both a driver of tumourigenesis and an accessory for immunosuppressive TGF-β signalling. Anti-tumour vaccines targeting p53 demonstrated significantly improved PFS when combined with chemotherapy among immunological OC responders [181]. An interventional trial is currently investigating the merit of a p53 vaccine in combination with pembrolizumab for recurrent OC patients [185].

Significantly improved survival outcomes of wild-type *BRCA* patients with the Vigil vaccine (an autologous tumour cell vaccine which downregulates TGF-β) justify phase III assessments and potential future FDA approval [197,198,199]. However, there are currently no immunotherapy-based clinical trials targeting either *KRAS* or *PTEN* mutated OC, which would be of value in the rarer OC subtypes. This is particularly pertinent given that these subtypes typically respond poorly to conventional platinum-based chemotherapy. Mutation of these genes is associated with activation of PI3K/AKT/mTOR pathway. Preclinical data support the use of PI3K inhibition to overcome ICB resistance [221]. In this regard, clinical trials investigating the PI3K inhibitor (copanlisib) alongside anti-PD-1 or anti-CTLA-4 therapies in solid tumours with *PTEN*/*PI3K* mutations are ongoing [222,223]. If successful, these could represent a novel combination for certain OC molecular subtypes.

Tumour-driven TME metabolic reprogramming restricts immune cell differentiation and function, such that targeting these pathways may prove key to enhance the efficacy of, and overcome resistance to, immunotherapy. Metformin reduces cellular respiration by inhibiting mitochondrial complex 1, limiting cancer cells’ metabolic plasticity as well as playing a central role in glucose and lipid metabolism. Preclinical studies on various solid tumour types and a retrospective analysis of melanoma clinical trial data suggest that metformin enhances CD8+ T-cell function and improves efficacy of ICB therapy [255,256,257]. Indeed, metformin treatment is associated with improved survival in diabetic patients with OC and is well-tolerated alongside standard chemotherapy (e.g., carboplatin/paclitaxel) in advanced OC [252,377]. Despite this, there are currently no OC clinical trials focussed on determining the efficacy of combining immunotherapy with metformin. As of yet, this notion is only being explored in the setting of both lung [261] and colorectal cancers [262].

Tumour cells escape hypoxia-induced apoptosis by activating the HIF pathway, which also contributes to establishing an immunosuppressive TME by recruiting intratumoural MDSCs and Tregs while upregulating tumour cell PD-L1 expression. Accordingly, targeting the HIF pathway may be one strategy to overcome ICB resistance. For example, a phase III trial investigating ICB in combination with VEGFR kinase inhibitors with or without a HIF-2α selective inhibitor is currently recruiting advanced renal cell cancer patients [315]. If results are encouraging, such an approach could potentially be translated to the OC setting.

Other immunosuppressive metabolic targets are currently being investigated. Recent preclinical research on the AhR receptor (which is activated by immunosuppressive metabolites of tryptophan produced by IDO1) demonstrated that AhR blockade can reverse TME immunosuppression and synergise with ICB to reduce tumour growth. The first trials to investigate the safety profile and dosing regimens of AhR inhibitors are ongoing [367], with two investigating the potential benefits of combination therapy with ICB in advanced solid tumours [368,369] although, disappointingly, they do not include OC. IL4I1 is a metabolic immune checkpoint that activates AhR through tryptophan metabolism and could thus constitute a resistance mechanism to IDO1-targeted therapies. IL4I1 is overexpressed in many cancers, including OC, and correlates with increased numbers of immunosuppressive cell TME infiltrates [370]. Elevated IL4I1 levels were noted in patients who received ICB therapy compared to ICB naïve patients, suggesting that it may have a role in the development of resistance to immunotherapy [370,378]. Thus, while IL4I1 represents a novel metabolic target with the potential to enhance the efficacy of immunotherapy, preclinical and clinical studies to understand its potential value in the OC setting remain wanting.

Many of the studies covered in this review highlight that the tantalising findings of preclinical research do not always bear out in clinical trials. In vitro tumour models are important tools for research and can be used for low-cost screening of drug therapies. However, they are often designed to replicate specific aspects of the TME rather than capturing its inherent complexity. Advances in three-dimensional tumour culture systems are increasingly focussed on replicating solid tumours’ complex in vivo histoarchitecture (e.g., involving tumour vasculature), as well as replicating the human immune system. This approach may allow future high-throughput drug screening and selection, thus accelerating candidate drug development programmes [379]. Moreover, most preclinical in vivo studies are conducted in mice, and efforts are currently under way to develop models that more closely reflect OC pathophysiology. For example, using oophorectomised mice better reflects the oestrogen levels of OC patients, most of whom are post-menopausal and/or have undergone oophorectomy as part of cytoreductive surgery. This is all the more relevant since oestrogen is known to impact immune cell function. In this regard, oophorectomised mouse models of HGSOC treated with PARPi in combination with anti-PD-1 therapy demonstrated a significant increase in memory CD8+ T-cells and CD4+/CD8+ T-cell IFN-γ production. This was associated with decreased tumour burden, thus emphasising the relevance of preclinical models accurately reflecting the hormonal TME established in OC [380]. Similarly, many mouse models are kept in a microbially defined environment, which may influence responses to therapy since the microbiome is known to affect immune behaviour. This is highlighted by evidence that co-housing laboratory mice with their pet shop counterparts upregulated both innate and adaptive pathways in the former, producing immune profiles closer to those of humans. Unfortunately, almost a quarter of the laboratory mice did not survive this exposure to an unscreened microbial environment [381]. Nevertheless, developing preclinical models to reflect the TME more accurately will enable the unlocking of the potential of immunotherapy regimens in OC.

Finally, most OC clinical trials of patients involving ICB therapy are conducted in women who have progressed despite conventional treatment. It is worth noting that these advanced tumours will likely have already established robust immune evasion strategies and consequently potentially have a lesser prognostic benefit in response to immunotherapy. Delayed adjuvant chemotherapy post cytoreductive surgery correlates with poorer survival, and it may be that earlier intervention with immunotherapy results in more favourable clinical outcomes. Developing a more detailed understanding of the integrated TME immunoregulatory, genomic and metabolic pathways involved in OC will be essential to enable us to develop, identify and target patients to effective, tailored treatments. In particular, targeted combination therapies addressing one or more of these areas may be needed to overcome the immunosuppressive TME in OC and maximise survival benefit. Patient stratification also has a role. Tailored treatment based on tumour genomic aberrations or PD-1/PD-L1 axis profiling will ascertain which OC patients have the greatest potential to derive survival advantages from immunotherapy. These efforts will highlight the value of immunotherapy in OC and increase clinicians’ support for its use and availability as part of a wider strategy to improve the durability of treatment response and survival.

## Figures and Tables

**Figure 1 cancers-13-06231-f001:**
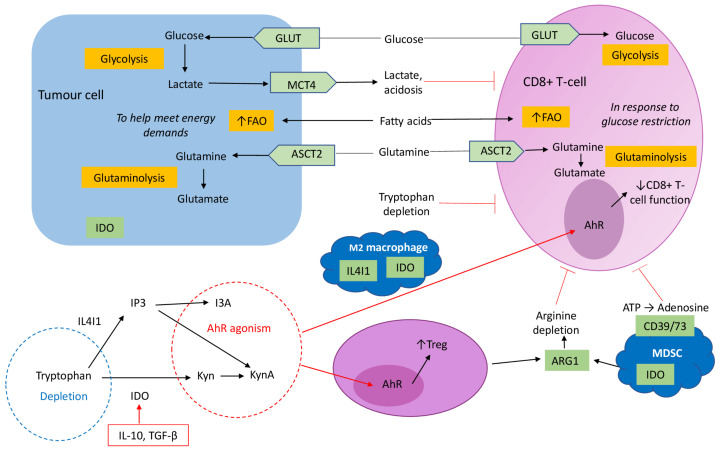
Effect of metabolic changes in the tumour microenvironment (TME) on immune cell differentiation. Increased tumour aerobic glycolysis, supported by upregulation of GLUT-mediated glucose provision provides energy for rapidly dividing cancer cells and leads to lactate accumulation within the TME. Fatty acid oxidation (FAO) is also utilised by tumour cells to supply energy for growth such that CD8+ T-cells starved of glucose increase FAO. Cancer cells also enhance amino acid metabolism to fulfil the energetic demands of rapid growth. Glutamine metabolism provides intermediates for the tricarboxylic acid cycle and maintains intracellular redox balance, making glutamine key to supporting cancer cell proliferation. This depletes glutamine from the TME, limiting its availability to CD8+ T-cells. Increased arginase 1 (ARG1) production by Tregs and MDSCs also limits arginine supply for CD8+ T-cell function. Cancer cell and M2 macrophage indolamine 2,3-dioxygenase (IDO) expression instead causes tryptophan depletion, which arrests CD8+ T-cell proliferation and upregulates regulatory T-cell (Treg) expansion. Tryptophan metabolism by IDO and IL4I1 yield the metabolites kynurenine (kyn), kynurenic acid (kyn A) and the indole I3a (indole-3-carboxaldehyde). Kyn and Kyn A are both ligands of the AhR receptor, which promote Treg differentiation and suppresses CD8+ T-cell function. Tumour transforming growth factor (TGF)-β induces upregulation of CD39/CD73 on MDSCs. These ectoenzymes hydrolyse adenosine triphosphate (ATP) to produce extracellular adenosine within the TME, which inhibits CD8+ T-cell proliferation. Abbreviations: GLUT: glucose transporters; MCT4: monocarboxylate transporter 4; ASCT2: glutamine transporter; IL-2/10: interleukin-2/10; TGF-β: transforming growth factor-β; IP3: indole-3-propionic acid; I3a: indole-3-carboxaldehyde; IL4I1: L-amino acid oxidase interleukin-4-induced-1.

**Figure 2 cancers-13-06231-f002:**
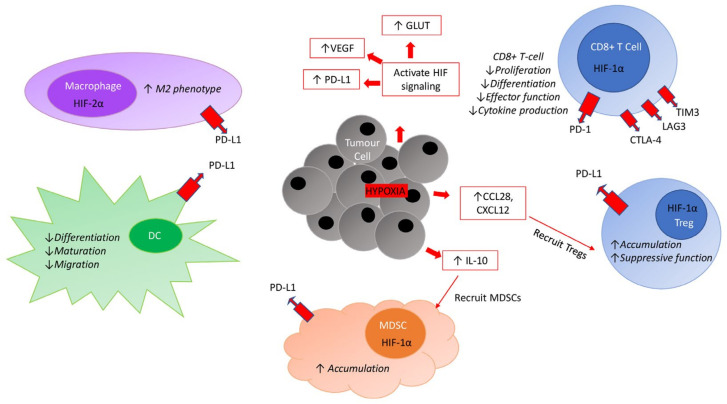
Hypoxia-driven immune escape within the tumour microenvironment (TME). The hypoxic environment promotes tumour production of interleukin (IL)-10 to recruit myeloid-derived suppressor cells (MDSCs) and regulatory T-cells (Tregs) to the TME. Hypoxia-induced expression of CC-chemokine ligand 28 (CCL28) and CXCL12 recruits Tregs to the TME. Hypoxia inducible factors (HIF)-1α and -2α transcriptionally upregulate vascular endothelial growth factors (VEGFs) to promote proangiogenic signalling in addition to recruiting Tregs to the TME. The HIF pathway also increases glucose transporters (GLUTs) to support substrate provision for tumour glycolysis. Upregulation of PD-L1 on tumour cells is HIF-1α-dependent. Impaired maturation and reduced production of cytokines such as interferon (IFN)-γ in dendritic cells (DCs) occurs in hypoxic environments. Upregulation of inhibitor programmed cell death ligand 1 (PD-L1) on DCs, Tregs, macrophages and MDSCs are all dependent on HIF signalling. TME Hypoxia regulates the expression of inhibitory checkpoint proteins. Cytotoxic T-lymphocyte antigen-4 (CTLA-4), lymphocyte activating 3 (LAG3), T-cell immunoglobulin domain and mucin domain 3 (TIM3) and PD-1 are up-regulated on CD8+ T-cells in a HIF-1α-dependent manner.

**Table 1 cancers-13-06231-t001:** Clinical trials in OC using immune checkpoint inhibition in combination with two other therapies. Advanced recurrent (AR), Platinum-resistant (PR), Recurrent or refractory (RR), Ovarian cancer (OC), Primary peritoneal cancer (PPC), Fallopian tube carcinoma (FTC), High-grade serous ovarian carcinoma (HGSOC).

Study Phase and Design	Inclusion Criteria	No. of Patients	Treatment	NCT (ClinicalTrials.gov)
Phase I/II, open-label, sequential assignment Recruiting	AR-PR OC, triple negative breast, lung, prostate or colorectal carcinoma	384	Durvalumab + olaparib +/− cediranib (anti-VEGF)	NCT02484404
Phase I/II, open-label, single group Active, not recruiting	RR OC/PPC/FTC with *BRCA 1/2* mutation	40	Olaparib + tremelimumab + durvalumab	NCT02953457
Phase II, triple masked, randomised Active, not recruiting	Recurrent PR OC/PPC/FTC	122	Atezolizumab + bevacizumab +/− placebo or acetylsalicylic acid	NCT02659384
Phase II, open-label, single group assignment Recruiting	Recurrent PR HGSOC	29	Atezolizumab + bevacizumab + cobimetinib (mitogen-activated protein kinase inhibitor)	NCT03363867
Phase II, open-label, single assignment Not yet recruiting	Recurrent PR OC/PPC/FTC or endometrial cancer	47	Lenvatinib + pembrolizumab + paclitaxel	NCT04781088
Phase II, open-labelled, randomised Not yet recruiting	Recurrent OC/PPC/FTC with *BRCA* wild-type	184	Maintainence post platinum chemotherapy of olaparib +/− durvalumab +/− UV1 vaccine (hTERT)	NCT04742075
Phase III, randomised, masked, parallel assignment Recruiting	Advanced epithelial OC with *BRCA* mutation	1284	First line treatment of carboplatin/paclitaxel + pembrolizumab or placebo Followed by maintenance of olaparib or placebo	NCT03740165
Phase III, randomized, double blinded, placebo controlled Active, not recruiting	Stage III/IV, high grade non-mucinous epithelial OC/PPC/FTC	1405	Carboplatin/paclitaxel + bevacizumab; + placebo Or + dostarlimab (anti-PD-1) Or + niraparib	NCT03602859
Phase III, randomized, double blinded, placebo controlled Active, not yet recruiting	Stage III/IV EOC/PPC/FTC who have completed cytoreductive surgery	1000	Maintenance post primary platinum-based chemotherapy; + nivolumab and rucaparib Or + nivolumab or placebo Or + rucaparib or placebo	NCT03522246
Phase III, randomised, double-blinded Recruiting	Recurrent high-grade serous or endometroid OC/PPC/FTC	414	Carboplatin/paclitaxel + atezolizumab or placebo Followed by niraparib maintenance + atezolizumab or placebo	NCT03598270
Phase III, randomised, parallel assignment Recruiting	Recurrent OC/PPC/FTC	664	Chemotherapy + bevacizumab + atezolizumab or placebo	NCT03353831
Phase III, randomised, parallel assignment Recruiting	Recurrent, high-grade, PR OC	444	Doxorubicin +/− atezolizumab +/− bevacizumab	NCT02839707
Phase III, randomised, double-blinded, placebo Recruiting	Advanced (III/IV) high-grade epithelial OC/PPC/FTC	1374	Platinum-based chemotherapy after primary/interval cytoreductive surgery and bevacizumab, followed by maintenance bevacizumab +/− durvalumab or placebo +/− olaparib or placebo	NCT03737643

**Table 2 cancers-13-06231-t002:** Chimaeric antigen receptor (CAR) T-cell therapy clinical trials in ovarian cancer.

Study Phase and Design	CAR Target	Eligible	No. of Patients	Treatment	NCT
Interventional open-label single group Recruiting	Anti-mesothelin	RR OC with mesothelin positive tumour	10 20	Cyclophosphamide + Fludarabine With CAR T-cells	NCT03814447
Interventional open-label single group Recruiting	Anti-mesothelin	RR OC with mesothelin positive tumour	20	CAR T-cells	NCT03916679
Phase 1 open-label single group Recruiting	Anti-mesothelin	RR OC with mesothelin positive tumour	34	Cyclophosphamide + Fludarabine With CAR T-cells	NCT04562298
Phase 1 open-label single group Recruiting	Anti-B7-H3 antigen	RR OC	21	Cyclophosphamide+ Fludarabine With CAR T-cells	NCT04670068
Phase 1 open-label single group Recruiting	Anti-MUC16 (gene encoding ca 125)	RR OC/PPC/FTC	71	Biological: PRGN-3005 UltraCAR T-cells	NCT03907527
Phase 1 open-label single group Not yet recruiting	Anti-ALPP	Metastatic ALPP positive OC and EC	20	CAR T-cells	NCT04627740
Phase 1 open-label single group Recruiting	Anti-α-FR	RR HGSOC/PPC/FTC with α-FR positive tumour	18	CAR T-cells with or without Cyclophosphamide + Fludarabine	NCT03585764
Exploratory open-label single group Recruiting	Anti-mesothelin T cells secreting PD-1 nanobodies	Mesothelin positive advanced solid tumours	10	CAR T-cells	NCT04503980
Interventional open-label single group Recruiting	Autogolous Immunogene-modified T-Cells (IgT)	Stage III/IV OC in complete remission post primary treatment	100	CAR T-cells	NCT03184753
Phase I open-label Recruiting	Anti-MUC1	Advanced MUC1+ solid tumours (refractory OC)	112	Cyclophosphamide + Fludarabine With CAR T-cells	NCT04025216

Relapsed refractory (RR), Ovarian cancer (OC), primary peritoneal cancer (PPC), Fallopian tube carcinoma (FTC), Endometrial cancer (EC), Alkaline phosphatase placental (ALPP), α-folate receptor (α-FR).

**Table 3 cancers-13-06231-t003:** Clinical trials in ovarian cancer using dendritic cell-based vaccines (DCVs).

Study Phase and Design	Eligible	No. of Patients	Controls	NCT
Phase I open-label single-armActive, not yet recruiting	IIIc/IV OC no residual disease post primary treatment	19	Folate receptor alpha loaded DCV only	NCT02111941
Phase I open-label single-armActive, not yet recruiting	HGSOC (=/>IIIb) post primary cytoreductive surgery + chemotherapy	17	DCV only	NCT04739527
Phase I/IIa open-label single-arm Active, not yet recruiting	Stage II-IV OC no residual disease post primary treatment	18	Alpha-type-1 polarised DCV and intra-peritoneal infusion of CTL	NCT03735589
Phase II Open-label single-arm Active, not yet recruiting	First recurrence of platinum-sensitive OC	33	Autologous maintenance DCV after standard chemotherapy	NCT03657966
Phase II open-label single-arm Recruiting	AR OC	36	Autologous DCV only loaded with tumour lysate or for patients who are HLA-A2 with peptides of MUC1 and *WT1* therapy	NCT00703105
Phase II Multicentre, randomised, double-blind, placebo-controlled Recruiting	Stage III/IV OC/PPC no residual disease post primary treatment	99	Autologous DCV only loaded with tumour antigen versus loaded with peripheral blood mononuclear Cells	NCT02033616
Phase II open-label randomised Active, not yet recruiting	AR OC	23	Autologous DCV plus GM-CSF	NCT00799110
Phase III Multicentre, randomised, double-blind, placebo-controlled Active, not yet recruiting	AR platinum-sensitive OC	678	Induction: DCV verus placebo with carboplatin + gemcitabine/paclitaxel/doxorubicin +/− bevacizumab Maintenance: DCV versus placebo + bevacizumab +/− PARPi	NCT03905902

Advanced recurrent (AR), Ovarian cancer (OC), Primary peritoneal cancer (PPC), Fallopian tube carcinoma (FTC), High-grade ovarian serous carcinoma (HGSOC), Wilms tumour 1 (*WT1*), Granulocyte–macrophage colony stimulating factor (GM-CSF), Poly (ADP-ribose) polymerase inhibitor (PARPi).

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
