# Peer review of "Barriers to Immunotherapy in Ovarian Cancer: Metabolic, Genomic, and Immune Perturbations in the Tumour Microenvironment"

_cancers, 2021, doi:10.3390/cancers13246231_

Round 1
Reviewer 1 Report
The Rewiev is well written and developed, especially from the point of view of the knowledge. However, in my opinion, it needs to be a little revised under the point of view of the paragraph structure to make the text more flowing and fluent. For instance the paragraph 2.1.1 is too long and it should be subdivided in more sub paragraphs. Also the last conclusion paragraph are too long and should be better summarized.
Author Response
Thank you. These points have been addressed and we trust that the new subcategorisation will offer greater clarity for the reader (please see amended document).
Reviewer 2 Report
Overall, this is a well-written, timely and comprehensive review.
Author Response
Thank you.
Reviewer 3 Report
The author in the current review, "Barriers to immunotherapy in ovarian cancer: metabolic, genomic, and immune perturbations in the tumour microenvironment", have extensively discussed all of the issues related to the ovarian cancer tumour microenvironment and the problem that challenges the efficacy of immunotherapy. They have discussed all modulating immunoregulatory pathways, reorchestrating metabolic pathways, and featuring specific cancer cell genomic aberrations.
Authors have done extremely good amount of work of this regards; however, I see there are more detailed in every single section that make it very hard to follow. Therefore, I suggest summarizing some of the sections will make the article readable for those who are interested to know details about this topic.
I suggest to accept the manuscript after proofreading and summarizing some of the contents.
Author Response
Thank you. In response, we have restructured the concluding section of the review in order to provide a more succinct summary of all the areas covered, highlighting the main take home messages. We have also proofread the manuscript, as requested.
Reviewer 4 Report
Dear Authors,
I have read the manuscript that is entitled as ''Barriers to immunotherapy in ovarian cancer: metabolic, genomic, and immune perturbations in the tumour microenvironment'' submitted to Cancers by Johnson RL et al.
They discussed specific TME features that prevent Immune Checkpoint Blackage (ICB) from reaching its full potential in immune, genomic and metabolic changes. They also gave the therapeutic strategies aiming to overcome these hurdles including clinical trials aim-ing to maximise the success of immunotherapy in OC.
I have read the manuscript with pleasure and I thank the authors to write such a great review.
Yours Sincerely,
Author Response
Thank you.
Reviewer 5 Report
This manuscript was well-written and provided novel insights in immunotherapy in ovarian cancer. The manuscript could be accepted in the present form.
Author Response
Thank you.